# AnomalyCLIP: Object-agnostic Prompt Learning for Zero-shot Anomaly Detection

**Qihang Zhou**[1*], **Guansong Pang**[2*], **Yu Tian**[3], **Shibo He**[1†], **Jiming Chen**[1†]
[1]Zhejiang University     [2] Singapore Management University     [3]Harvard University
[1]{zqhang, s18he, cjm}@zju.edu.cn   [2]gspang@smu.edu.sg
[3]ytian11@meei.harvard.edu

## Abstract

Zero-shot anomaly detection (ZSAD) requires detection models trained using auxiliary data to detect anomalies without any training sample in a target dataset. It is a crucial task when training data is not accessible due to various concerns, *e.g.*, data privacy, yet it is challenging since the models need to generalize to anomalies across different domains where the appearance of foreground objects, abnormal regions, and background features, such as defects/tumors on different products/organs, can vary significantly. Recently large pre-trained vision-language models (VLMs), such as CLIP, have demonstrated strong zero-shot recognition ability in various vision tasks, including anomaly detection. However, their ZSAD performance is weak since the VLMs focus more on modeling the class semantics of the foreground objects rather than the abnormality/normality in the images. In this paper we introduce a novel approach, namely AnomalyCLIP, to adapt CLIP for accurate ZSAD across different domains. The key insight of AnomalyCLIP is to learn object-agnostic text prompts that capture generic normality and abnormality in an image regardless of its foreground objects. This allows our model to focus on the abnormal image regions rather than the object semantics, enabling generalized normality and abnormality recognition on diverse types of objects. Large-scale experiments on 17 real-world anomaly detection datasets show that AnomalyCLIP achieves superior zero-shot performance of detecting and segmenting anomalies in datasets of highly diverse class semantics from various defect inspection and medical imaging domains. Code will be made available at https://github.com/zqhang/AnomalyCLIP.

## 1 Introduction

Anomaly detection (AD) has been widely applied in various applications, such as industrial defect inspection (Bergmann et al., 2019; 2020; Liznerski et al., 2020; Pang et al., 2021a; Roth et al., 2022; Huang et al., 2022; Mou et al., 2022; Chen et al., 2022; You et al., 2022; Ding et al., 2022; Reiss & Hoshen, 2023; Xie et al., 2023; Zhou et al., 2023; Cao et al., 2023) and medical image analysis (Pang et al., 2021a; Tian et al., 2021; Fernando et al., 2021; Qin et al., 2022; Ding et al., 2022; Liu et al., 2023; Tian et al., 2023). Existing AD approaches typically assume that training examples in a target application domain are available for learning the detection models (Pang et al., 2021b; Ruff et al., 2021). However, this assumption may not hold in various scenarios, such as i) when accessing training data violates data privacy policies (*e.g.*, to protect the sensitive information of patients), or ii) when the target domain does not have relevant training data (*e.g.*, inspecting defects in a manufacturing line of new products). Zero-shot anomaly detection (ZSAD) is an emerging task for AD in such scenarios, to which the aforementioned AD approaches are not viable, as it requires detection models to detect anomalies without any training sample in a target dataset.

Since anomalies from different application scenarios typically have substantial variations in their visual appearance, foreground objects, and background features, *e.g.*, defects on the surface of one product vs. that on the other products, lesions/tumors on different organs, or industrial defects vs. tumors/lesions in medical images, detection models with strong generalization ability w.r.t. such variations are needed for accurate ZSAD. Recently large pre-trained vision-language models (VLMs) (Radford et al., 2021; Kirillov et al., 2023) have demonstrated strong zero-shot recognition ability in various vision tasks, including anomaly detection (Jeong et al., 2023). Particularly, being

---

*Equal contribution. † Corresponding authors.

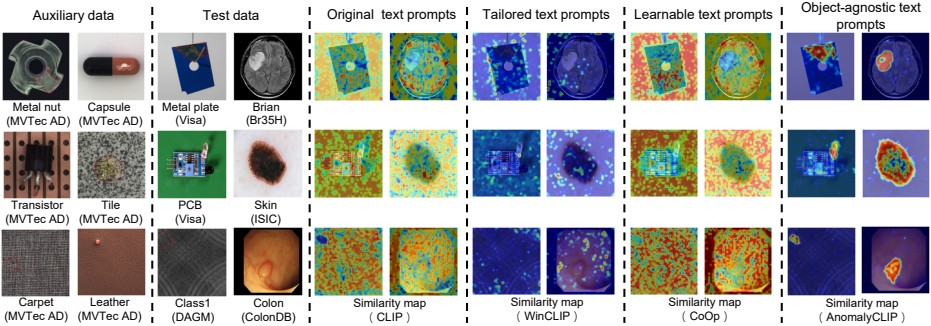

Figure 1: Comparison of ZSAD results on (b) test data using (c) original text prompts in CLIP (Radford et al., 2021), (d) tailored text prompts for AD in WinCLIP (Jeong et al., 2023), (e) learnable text prompts for general vision tasks in CoOp (Zhou et al., 2022a), and (f) object-agnostic text prompts in our AnomalyCLIP. (a) presents a set of auxiliary data we can use to learn the text prompts. The results are obtained by measuring the similarity between text prompt embeddings and image embeddings. The ground-truth anomaly regions are circled in red in (a) and (b). (c), (d), and (e) suffer from poor generalization across different domains, while our AnomalyCLIP in (f) can well generalize to anomalies in diverse types of objects from different domains.

pre-trained using millions/billions of image-text pairs, CLIP (Radford et al., 2021) has been applied to empower various downstream tasks (Zhou et al., 2022b; Rao et al., 2022; Khattak et al., 2023; Sain et al., 2023) with its strong generalization capability. WinCLIP (Jeong et al., 2023) is a seminal work in the ZSAD line, which designs a large number of artificial text prompts to exploit the CLIP's generalizability for ZSAD. However, the VLMs such as CLIP are primarily trained to align with the class semantics of foreground objects rather than the abnormality/normality in the images, and as a result, their generalization in understanding the visual abnormality/normality is restricted, leading to weak ZSAD performance. Further, the current prompting approaches, using either manually defined text prompts (Jeong et al., 2023) or learnable prompts (Sun et al., 2022; Zhou et al., 2022a), often result in prompt embeddings that opt for global features for effective object semantic alignment (Zhong et al., 2022; Wu et al., 2023), failing to capture the abnormality that often manifests in fine-grained, local features, as shown in Fig. 1d and Fig. 1e.

In this paper we introduce a novel approach, namely AnomalyCLIP, to adapt CLIP for accurate ZSAD across different domains. AnomalyCLIP aims to learn object-agnostic text prompts that capture generic normality and abnormality in an image regardless of its foreground objects. It first devises a simple yet universally-effective learnable prompt template for the two general classes – normality and abnormality – and then utilizes both image-level and pixel-level loss functions to learn the generic normality and abnormality globally and locally in our prompt embeddings using auxiliary data. This allows our model to focus on the abnormal image regions rather the object semantics, enabling remarkable zero-shot capability of recognizing the abnormality that has similar abnormal patterns to those in auxiliary data. As shown in Fig. 1a and Fig. 1b, the foreground object semantics can be completely different in the fine-tuning auxiliary data and target data, but the anomaly patterns remain similar, *e.g.*, scratches on metal nuts and plates, the misplacement of transistors and PCB, tumors/lesions on various organ surfaces, etc. Text prompt embeddings in CLIP fail to generalize across different domains, as illustrated in Fig. 1c, but object-agnostic prompt embeddings learned by AnomalyCLIP can effectively generalize to recognize the abnormality across different domain images in Fig. 1f. In summary, this paper makes the following main contributions.

- We reveal for the first time that learning object-agnostic text prompts of normality and abnormality is a simple yet effective approach for accurate ZSAD. Compared to current text prompting approaches that are primarily designed for object semantic alignment (Zhou et al., 2022b; Jeong et al., 2023), our text prompt embeddings model semantics of generic abnormality and normality, allowing object-agnostic, generalized ZSAD performance.

- We then introduce a novel ZSAD approach, called AnomalyCLIP, in which we utilize an object-agnostic prompt template and a glocal abnormality loss function (i.e., a combination of global and local loss functions) to learn the generic abnormality and normality prompts using auxiliary data. In doing so, AnomalyCLIP largely simplifies the prompt design and can effectively apply to different domains without requiring any change on its learned two

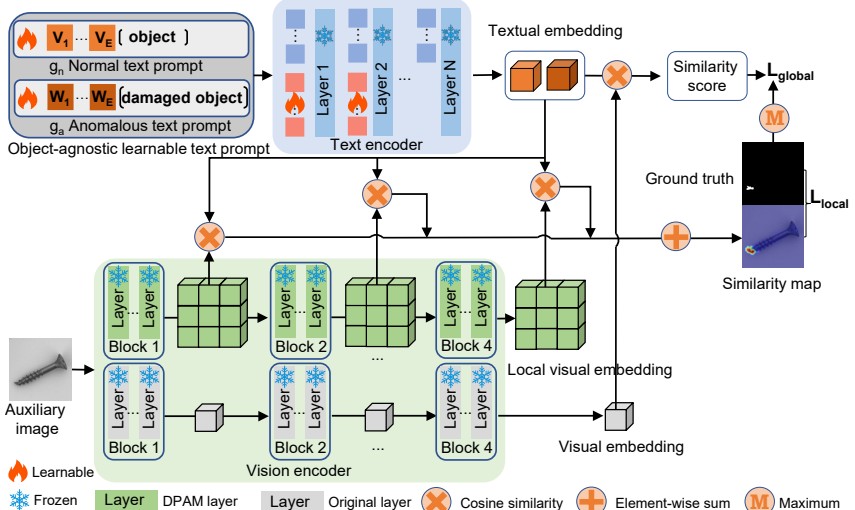

Figure 2: Overview of AnomalyCLIP. To adapt CLIP to ZSAD, AnomalyCLIP introduces object-agnostic text prompt templates to capture generic normality and abnormality regardless of the object semantics. Then, we introduce glocal context optimization to incorporate global and fine-grained anomaly semantics into object-agnostic text prompt learning. Finally, textual prompt tuning and DPAM are used to enable the prompt learning in the textual and local visual spaces of CLIP.

> prompts, contrasting to existing methods like WinCLIP whose effectiveness relies heavily on extensive engineering on hundreds of manually defined prompts.

- Comprehensive experiments on 17 datasets from various industrial and medical domains demonstrate that AnomalyCLIP achieves superior ZSAD performance of detecting and segmenting anomalies in datasets of highly diverse class semantics from defect inspection and medical imaging domains.

## 2 PRELIMINARY

CLIP consists of a text encoder and visual encoder denoted as $T(\cdot)$ and $F(\cdot)$, respectively. Both encoders are mainstream multi-layer networks such as ViT (Vaswani et al., 2017; Dosovitskiy et al., 2020). Using text prompts is a typical way to achieve the embeddings of different classes for zero-shot recognition. Particularly, a text prompt template $\mathbb{G}$ with the class name $c$ can be passed through $T(\cdot)$ to obtain its corresponding textual embedding $g_c \in \mathbb{R}^D$. The text prompt template commonly used in CLIP looks like `A photo of a [cls]`, where `[cls]` represents the target class name. Then $F(\cdot)$ encodes an image $x_i$ to derive visual representations, where the class token $f_i \in \mathbb{R}^D$ is treated as its visual embedding (global visual embedding), and patch tokens $f_i^m \in \mathbb{R}^{H \times W \times D}$ are referred to as local visual embeddings. CLIP performs zero-shot recognition by measuring the similarity between textual and visual embeddings. In specific, given a target class set $\mathcal{C}$ and an image $x_i$, CLIP predicts the probability of $x_i$ belonging to $c$ as follows:

$$p(y = c|x_i) = P(g_c, f_i) = \frac{exp(<g_c, f_i>/\tau)}{\sum_{c \in \mathcal{C}} exp(<g_c, f_i>)/\tau)}, \quad (1)$$

where $\tau$ is a temperature hyperparameter, and the operator $< \cdot, \cdot >$ represents the computation of cosine similarity. Unlike many vision tasks that involve many objects and use the name of the objects as the class name `[cls]`, we posit that performing ZSAD tasks using CLIP should be object-agnostic, so we propose to design two classes of text prompts (i.e., normality and abnormality) and compute the possibility of these two classes according to Eq. 1. We denote the probability of being abnormal $P(g_a, f_i)$ as the anomaly score. The computation is extended from global visual embeddings to local visual embeddings to derive the corresponding segmentation maps $S_n \in \mathbb{R}^{H \times W}$ and $S_a \in \mathbb{R}^{H \times W}$, where each entry $(j, k)$ are computed as $P(g_n, f_i^{m(j,k)})$ and $P(g_a, f_i^{m(j,k)})$.

## 3 ANOMALYCLIP: OBJECT-AGNOSTIC PROMPT LEARNING

### 3.1 APPROACH OVERVIEW

In this paper, we propose AnomalyCLIP to adapt CLIP to ZSAD via object-agnostic prompt learning. As shown in Fig. 2, AnomalyCLIP first introduces object-agnostic text prompt templates, where

we design two generic object-agnostic text prompt templates of $g_n$ and $g_a$ to learn generalized embedding for the normality and abnormality classes, respectively (see Sec. 3.2). To learn such generic text prompt templates, we introduce global and local context optimization to incorporate global and fine-grained anomaly semantics into object-agnostic textual embedding learning. In addition, textual prompt tuning and DPAM are used to support the learning in the textual and local visual spaces of CLIP. Finally, we integrate the multiple intermediate layers to provide more local visual details. During training, all modules are jointly optimized by the combination of global and local context optimization. During inference, we quantify the misalignment of textual and global/local visual embeddings to obtain the anomaly score and anomaly score map, respectively (see Sec. 3.3).

## 3.2 OBJECT-AGNOSTIC TEXT PROMPT DESIGN

Commonly used text prompt templates in CLIP, like `A photo of a [cls]`, primarily focus on object semantics. Consequently, they fail to generate textual embeddings that capture anomaly and normal semantics to query corresponding visual embeddings. To support the learning of anomaly-discriminative textual embeddings, we aim to incorporate prior anomaly semantics into text prompt templates. A trivial solution is to design the templates with specific anomaly types, such as `A photo of a [cls] with scratches`. However, the pattern of anomaly is typically unknown and diverse, so it is practically difficult to list all possible anomaly types. Therefore, it is important to define text prompt templates with generic anomaly semantics. For this purpose, we can adopt the text `damaged [cls]` to cover comprehensive anomaly semantics, facilitating the detection of diverse defects such as scratches and holes. Nevertheless, utilizing such text prompt templates poses challenges in generating generic anomaly-discriminating textual embeddings. This is because CLIP's original pre-training focuses on aligning with object semantics instead of the abnormality and normality within images. To address this limitation, we can introduce learnable text prompt templates and tune the prompts using auxiliary AD-relevant data. During the fine-tuning process, these learnable templates can incorporate both broad and detailed anomaly semantics, resulting in textual embeddings that are more discriminative between normality and abnormality. This helps avoid the need for manually defined text prompt templates that require extensive engineering (Jeong et al., 2023). These text prompts are referred to as **object-aware text prompt templates** and defined as follows:

$$g_n = [V_1][V_2] \ldots [V_E][cls]$$
$$g_a = [W_1][W_2] \ldots [W_E][damaged][cls],$$

where $[V]_i$ and $[W]_i$ ($i \in \{1, \ldots, E\}$) are learnable word embeddings in normality and abnormality text prompt templates, respectively.

ZSAD tasks require models to detect anomalies in previously unseen target datasets. These datasets often exhibit significant variations in object semantics among different objects, like various defects on one product vs. another, or discrepancies between industrial defects and medical imaging tumors. However, despite these substantial differences in object semantics, the underlying anomaly patterns could be similar. For instance, anomalies like scratches on metal nuts and plates, or the misplacement of transistors and PCB, as well as tumors on the surface of various organs, can share similar anomaly patterns. We hypothesize that the key of accurate ZSAD is to identify these generic anomaly patterns regardless of the varying semantics of different objects. Therefore, the inclusion of object semantics in object-aware text prompt templates is often unnecessary for ZSAD. It can even hinder the detection of anomalies in classes that have not been seen during the learning process. More importantly, excluding the object semantics from text prompt templates allows learnable text prompt templates to focus on capturing the characteristics of anomalies themselves, rather than the objects. Motivated by this, we introduce object-agnostic prompt learning, with the aim to capture generic normality and abnormality within images regardless of the object semantics. Different from object-aware text prompt templates, as shown below, the **object-agnostic text prompt templates** replace the class name in $g_n$ and $g_a$ with `object`, blocking out the class semantics of objects:

$$g_n = [V_1][V_2] \ldots [V_E][object]$$
$$g_a = [W_1][W_2] \ldots [W_E][damaged][object].$$

This design empowers the object-agnostic text prompt template to learn the shared patterns of different anomalies. As a result, the generated textual embeddings are more generic and capable of identifying anomalies across diverse objects and different domains. Further, this prompt design is versatile and can be applied to different target domains without any modification, *e.g.*, requiring no knowledge about the object name or anomaly types in a target dataset.

### 3.3 LEARNING GENERIC ABNORMALITY AND NORMALITY PROMPTS

**Glocal context optimization**   To effectively learn the object-agnostic text prompts, we devise a joint optimization approach that enables the normality and abnormality prompt learning from both global and local perspectives, namely global and local context optimization. The global context optimization aims to enforce that our object-agnostic textual embeddings are matched with the global visual embeddings of images of diverse objects. This helps effectively capture the normal/abnormal semantics from a global feature perspective. The local context optimization is introduced to enable object-agnostic text prompts to concentrate on fine-grained, local abnormal regions from $M$ intermediate layers of the visual encoder, in addition to the global normal/abnormal features. Formally, let $\mathcal{M}$ be the set of intermediate layers used (*i.e.*, $M = |\mathcal{M}|$), our text prompts are learned by minimizing the following glocal loss function:

$$L_{total} = L_{global} + \lambda \sum_{M_c \in \mathcal{M}} L_{local}^{M_c}, \tag{2}$$

where $\lambda$ is a hyperparameter to balance the global and local losses. $L_{global}$ is a cross-entropy loss that matches the cosine similarity between the object-agnostic textual embeddings and visual embeddings of normal/abnormal images from auxiliary data. Let $S \in \mathbb{R}^{H_{image} \times W_{image}}$ be the ground-truth segmentation mask, with $S_{jk} = 1$ if the pixel is as an anomaly and $S_{jk} = 0$ otherwise, then we have

$$S_{n,M_c}^{(j,k)} = P(g_n, f_{i,M_c}^{m(j,k)}), \;\; S_{a,M_c}^{(j,k)} = P(g_a, f_{i,M_c}^{m(j,k)}), \;\; \text{where} \;\; j \in [1,H], k \in [1,W]$$

$$L_{local} = Focal(Up([S_{n,M_c}, S_{a,M_c}]), S) + Dice(Up(S_{n,M_c}), I - S) + Dice(Up(S_{a,M_c}), S),$$

where $Focal(\cdot, \cdot)$ and $Dice(\cdot, \cdot)$ denote a focal loss (Lin et al., 2017) and a Dice loss (Li et al., 2019) respectively. The operators $Up(\cdot)$ and $[\cdot, \cdot]$ represent the unsampling and concatenation along with the channel, and $I$ represents the full-one matrix. Since the anomalous regions are typically smaller than the normal ones, we use focal loss to address the imbalance problem. Furthermore, to ensure that the model establishes an accurate decision boundary, we employ the Dice loss to measure the overlaps between the predicted segmentation $Up(S_{n,M_c})/Up(S_{a,M_c})$ and the ground truth mask.

**Refinement of the textual space**   To facilitate the learning of a more discriminative textual space via Eq. 2, inspired by Jia et al. (2022) and Khattak et al. (2023), we introduce text prompt tuning to refine the original textual space of CLIP by adding additional learnable token embeddings into its text encoder. Specifically, we first attach randomly initialized learnable token embeddings $t'_m$ into $T_m$, the $m$-th layer of the frozen CLIP text encoder. Then, we concatenate $t'_m$ and the original token embeddings $t_m$ along the dimension of the channel, and forward them to $T_m$ to get the corresponding $r'_{m+1}$ and $t_{m+1}$. To ensure proper calibration, we discard the obtained $r'_{m+1}$ and initialize new learnable token embeddings $t'_{m+1}$. Note that even though the output $r'_{m+1}$ is discarded, the updated gradients can still be backpropagated to optimize the learnable tokens $t'_m$ due to the self-attention mechanism. We repeat this operation until we reach the designated layer $M'$. During fine-tuning, these learnable token embeddings are optimized to refine the original textual space. More details see Appendix D.

**Refinement of the local visual space**   Since the visual encoder of CLIP is originally pre-trained to align global object semantics, the contrastive loss used in CLIP makes the visual encoder produce a representative global embedding for class recognition. Through the self-attention mechanism, the attention map in the visual encoder focuses on the specific tokens highlighted within the red rectangle in Fig 3b. Although these tokens may contribute to global object recognition, they disrupt the local visual semantics, which directly hinders the effective learning of the fine-grained abnormality in our object-agnostic text prompts (Li et al., 2023b). We found empirically that a

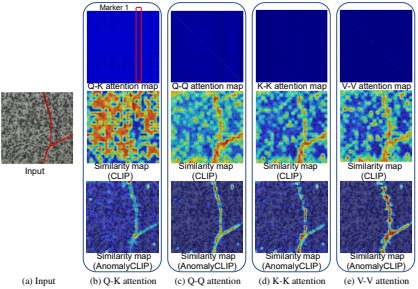

Figure 3: DPAM visualization.

diagonally prominent attention map helps reduce the disturbance from other tokens, leading to improved local visual semantics. Therefore, we propose a mechanism called Diagonally Prominent Attention Map to refine the local visual space, with the visual encoder kept frozen during training. To this end, we replace the original $Q$-$K$ attention in the visual encoder with diagonally prominent attention, such as $Q$-$Q$, $K$-$K$, and $V$-$V$ self-attention schemes. As demonstrated in Fig.3c, Fig.3d, and Fig. 3e, the refined DPAM attention maps are more diagonally prominent, resulting in substantially improved segmentation maps in both original CLIP and our AnomalyCLIP. Compared to

CLIP that is based on global features and manually defined text prompts, the text prompts learned by AnomalyCLIP are more fine-grained, enabling substantially more accurate alignment between the normality/abnormality prompt embeddings and the local visual embeddings across four different self-attention schemes. This, in turn, allows AnomalyCLIP to generate accurate $S_n$ and $S_a$ for the joint optimization in Eq. 2. Unless otherwise specified, AnomalyCLIP utilizes $V$-$V$ self-attention due to its superior overall performance. The performance of different self-attention mechanisms is analyzed in Sec. D. We also provide a detailed explanation about DPAM in Appendix C.

**Training and Inference** During training, AnomalyCLIP minimizes the loss in Eq. 2 using an auxiliary AD-related dataset. As for inference, given a test image $x_i$, we use the similarity score $P(g_a, f_i)$ as the image-level anomaly score, with the anomaly score leaning toward one when the anomaly textual embedding $g_a$ is aligned with global visual embedding $f_i$. For pixel-wise predictions, we merge the segmentation $S_{n,M_c}$ and $S_{a,M_c}$ of all selected intermediate layers, followed by an interpolation and smoothing operation. Formally, our anomaly score map $Map \in \mathbb{R}^{H_{image} \times W_{image}}$ is computed as $Map = G_\sigma(\sum_{M_c \in \mathcal{M}}(\frac{1}{2}(I - Up(S_{n,M_c})) + \frac{1}{2}Up(S_{a,M_c})))$, where $G_\sigma$ represents a Gaussian filter, and $\sigma$ controls smoothing.

## 4 EXPERIMENTS

### 4.1 EXPERIMENT SETUP

**Datasets and Evaluation Metrics** We conducted extensive experiments on 17 publicly available datasets, covering various industrial inspection scenarios and medical imaging domains (including photography, endoscopy, and radiology) to evaluate the performance of AnomalyCLIP. In industrial inspection, we consider MVTec AD (Bergmann et al., 2019), VisA (Zou et al., 2022), MPDD (Jezek et al., 2021), BTAD (Mishra et al., 2021), SDD (Tabernik et al., 2020), DAGM (Wieler & Hahn, 2007), and DTD-Synthetic (Aota et al., 2023). In medical imaging, we consider skin cancer detection dataset ISIC (Gutman et al., 2016), colon polyp detection datasets CVC-ClinicDB (Bernal et al., 2015), CVC-ColonDB (Tajbakhsh et al., 2015), Kvasir (Jha et al., 2020), and Endo (Hicks et al., 2021), thyroid nodule detection dataset TN3k (Gong et al., 2021), brain tumor detection datasets HeadCT (Salehi et al., 2021), BrainMRI (Salehi et al., 2021), Br35H (Hamada., 2020), and COVID-19 detection dataset COVID-19 (Chowdhury et al., 2020; Rahman et al., 2021). The SOTA competing methods include CLIP (Radford et al., 2021), CLIP-AC (Radford et al., 2021), WinCLIP (Jeong et al., 2023), VAND (Chen et al., 2023), and CoOp (Zhou et al., 2022b). We provide more details about the methods and data pre-processing in Appendix A. The anomaly detection performance is evaluated using the Area Under the Receiver Operating Characteristic Curve (AUROC). Additionally, average precision (AP) for anomaly detection and AUPRO (Bergmann et al., 2020) for anomaly segmentation are also used to provide more in-depth analysis of the performance.

**Implementation details** We use the publicly available CLIP model[1] (`VIT-L/14@336px`) as our backbone. Model parameters of CLIP are all frozen. The length of learnable word embeddings $E$ is set to 12. The learnable token embeddings are attached to the first 9 layers of the text encoder for refining the textual space, and their length in each layer is set to 4. We fine-tune AnomalyCLIP using the test data on MVTec AD and evaluate the ZSAD performance on other datasets. As for MVTec AD, we fine-tune AomalyCLIP on the test data of VisA. We report dataset-level results, which are averaged across their respective sub-datasets. All experiments are conducted in PyTorch-2.0.0 with a single NVIDIA RTX 3090 24GB GPU. More details can be found in Appendix A.

### 4.2 MAIN RESULTS

**ZSAD performance on diverse industrial inspection domains** Table 1 shows the ZSAD results of AnomalyCLIP with five competing methods over seven industrial defect datasets of very different foreground objects, background, and/or anomaly types. AnomalyCLIP achieves superior ZSAD performance across the datasets, substantially outperforming the other five methods in most datasets. The weak performance of CLIP and CLIP-AC can be attributed to CLIP's original pre-training, which focuses on aligning object semantics rather than anomaly semantics. By using manually defined text prompts, WinCLIP and VAND achieve better results. Alternatively, CoOp adopts learnable prompts to learn the global anomaly semantics. However, those prompts focus on the global feature and ignore the fine-grained local anomaly semantics, leading to their poor performance on anomaly segmentation. To adapt CLIP to ZSAD, AnomalyCLIP learns object-agnostic text prompts to focus on learning the generic abnormality/normality using global and local context optimization,

---

[1]https://github.com/mlfoundations/open_clip

Table 1: ZSAD performance comparison on industrial domain. The best performance is highlighted in red, and the second-best is highlighted in blue. † denotes results taken from original papers.

| Task | Category | Datasets | $|\mathcal{C}|$ | CLIP | CLIP-AC | WinCLIP | VAND | CoOp | AnomalyCLIP |
|---|---|---|---|---|---|---|---|---|---|
| Image-level (AUROC, AP) | Obj &texture | MVTec AD | 15 | (74.1, 87.6) | (71.5, 86.4) | (91.8, 96.5)† | (86.1, 93.5)† | (88.8, 94.8) | (91.5, 96.2) |
| | Obj | VisA | 12 | (66.4, 71.5) | (65.0, 70.1) | (78.1, 81.2)† | (78.0, 81.4)† | (62.8, 68.1) | (82.1, 85.4) |
| | | MPDD | 6 | (54.3, 65.4) | (56.2, 66.0) | (63.6, 69.9) | (73.0, 80.2) | (55.1, 64.2) | (77.0, 82.0) |
| | | BTAD | 3 | (34.5, 52.5) | (51.0, 62.1) | (68.2, 70.9) | (73.6, 68.6) | (66.8, 77.4) | (88.3, 87.3) |
| | | SDD | 1 | (65.7, 45.2) | (65.2, 45.7) | (84.3, 77.4) | (79.8, 71.4) | (74.9, 65.1) | (84.7, 80.0) |
| | Texture | DAGM | 10 | (79.6, 59.0) | (82.5, 63.7) | (91.8, 79.5) | (94.4, 83.8) | (87.5, 74.6) | (97.5, 92.3) |
| | | DTD-Synthetic | 12 | (71.6, 85.7) | (66.8, 83.2) | (93.2, 92.6) | (86.4, 95.0) | (-, -) | (93.5, 97.0) |
| Pixel-level (AUROC, PRO) | Obj &texture | MVTec AD | 15 | (38.4, 11.3) | (38.2, 11.6) | (85.1, 64.6)† | (87.6, 44.0)† | (33.3, 6.7) | (91.1, 81.4) |
| | Obj | VisA | 12 | (46.6, 14.8) | (47.8, 17.3) | (79.6, 56.8)† | (94.2, 86.8)† | (24.2, 3.8) | (95.5, 87.0) |
| | | MPDD | 6 | (62.1, 33.0) | (58.7, 29.1) | (76.4, 48.9) | (94.1, 83.2) | (15.4, 2.3) | (96.5, 88.7) |
| | | BTAD | 3 | (30.6, 4.4) | (32.8, 8.3) | (72.7, 27.3) | (60.8, 25.0) | (28.6, 3.8) | (94.2, 74.8) |
| | | SDD | 1 | (39.0, 8.9) | (32.5, 5.8) | (68.8, 24.2) | (79.8, 65.1) | (28.9, 7.1) | (90.6, 67.8) |
| | Texture | DAGM | 10 | (28.2, 2.9) | (32.7, 4.8) | (87.6, 65.7) | (82.4, 66.2) | (17.5, 2.1) | (95.6, 91.0) |
| | | DTD-Synthetic | 12 | (33.9, 12.5) | (23.7, 5.5) | (83.9, 57.8) | (95.3, 86.9) | (-, -) | (97.9, 92.3) |

Table 2: ZSAD performance comparison on medical domain. The best performance is highlighted in red, and the second-best is highlighted in blue. Note that the image-level medical AD datasets do not contain segmentation ground truth, so the pixel-level medical AD datasets are different from the image-level datasets.

| Task | Category | Datasets | $|\mathcal{C}|$ | CLIP | CLIP-AC | WinCLIP | VAND | CoOp | AnomalyCLIP |
|---|---|---|---|---|---|---|---|---|---|
| Image-level (AUROC, AP) | Brain | HeadCT | 1 | (56.5, 58.4) | (60.0, 60.7) | (81.8, 80.2) | (89.1, 89.4) | (78.4, 78.8) | (93.4, 91.6) |
| | | BrainMRI | 1 | (73.9, 81.7) | (80.6, 86.4) | (86.6, 91.5) | (89.3, 90.9) | (61.3, 44.9) | (90.3, 92.2) |
| | | Br35H | 1 | (78.4, 78.8) | (82.7, 81.3) | (80.5, 82.2) | (93.1, 92.9) | (86.0, 87.5) | (94.6, 94.7) |
| | Chest | COVID-19 | 1 | (73.7, 42.4) | (75.0, 45.9) | (66.4, 42.9) | (15.5, 8.5) | (25.3, 9.2) | (80.1, 58.7) |
| Pixel-level (AUROC, PRO) | Skin | ISIC | 1 | (33.1, 5.8) | (36.0, 7.7) | (83.3, 55.1) | (89.4, 77.2) | (51.7, 15.9) | (89.7, 78.4) |
| | Colon | CVC-ColonDB | 1 | (49.5, 15.8) | (49.5, 11.5) | (70.3, 32.5) | (78.4, 64.6) | (40.5, 2.6) | (81.9, 71.3) |
| | | CVC-ClinicDB | 1 | (47.5, 18.9) | (48.5, 12.6) | (51.2, 13.8) | (80.5, 60.7) | (34.8, 2.4) | (82.9, 67.8) |
| | | Kvasir | 1 | (44.6, 17.7) | (45.0, 16.8) | (69.7, 24.5) | (75.0, 36.2) | (44.1, 3.5) | (78.9, 45.6) |
| | | Endo | 1 | (45.2, 15.9) | (46.6, 12.6) | (68.2, 28.3) | (81.9, 54.9) | (40.6, 3.9) | (84.1, 63.6) |
| | Thyroid | TN3K | 1 | (42.3, 7.3) | (35.6, 5.2) | (70.7, 39.8) | (73.6, 37.8) | (34.0, 9.5) | (81.5, 50.4) |

enabling the modeling of both global and local abnormality/normality. Our resulting prompts can also generalize to different datasets from various domains. To provide more intuitive results, we visualize the anomaly segmentation results of AnomalyCLIP, VAND, and WinCLIP across different datasets in Fig. 4. Compared to VAND and WinCLIP, AnomalyCLIP can perform much more accurate segmentation for the defects from different industrial inspection domains.

**Generalization from defect datasets to diverse medical domain datasets** To evaluate the generalization ability of our model, we further examine the ZSAD performance of AnomalyCLIP on 10 medical image datasets of different organs across different imaging devices. Table 2 shows the results, where learning-based methods, including AnomalyCLIP, VAND and CoOp, are all tuned using MVTec AD data. It is remarkable that methods like AnomalyCLIP and VAND obtain promising ZSAD performance on various medical image datasets, even though they are tuned using a defect detection dataset. Among all these methods, AnomalyCLIP is the best performer due to its strong generalization brought by object-agnostic prompt learning. As illustrated in Fig. 4, AnomalyCLIP can accurately detect various types of anomalies in diverse medical images, such as skin cancer regions in photography images, colon polyps in endoscopy images, thyroid nodules in ultrasound images, and brain tumors in MRI images, having substantially better performance in locating the abnormal lesion/tumor regions than the other two methods WinCLIP and VAND. This again demonstrates the superior ZSAD performance of AnomalyCLIP in datasets of highly diverse object semantics from medical imaging domains.

**Can we obtain better ZSAD performance if fine-tuned using medical image data?** Comparing the promising performance in industrial datasets, AnomalyCLIP presents a relatively low performance in medical datasets. This is partly due to the impact of auxiliary data used in our prompt learning. So, then we examine whether the ZSAD performance on medical images can be improved if the prompt learning is trained on an auxiliary medical dataset. One challenge is that there are no available large 2D medical datasets that include both image-level and pixel-level annotations for our training. To address this issue, we create such a dataset based on ColonDB (More details see Appendix A), and then optimize the prompts in AnomalyCLIP and VAND using this dataset and evaluate their performance on the medical image datasets. The results are presented in Table 3. AnomalyCLIP and VAND largely improve their detection and segmentation performance compared to that fine-tuned on MVTec AD, especially for the colon polyp-related datasets such as CVC-ClincDB, Kvasir, and Endo (note that these datasets are all from different domains compared to the fine-tuning ColonDB dataset). AnomalyCLIP also exhibits performance improvement in detecting

brain tumors in datasets such as HeadCT, BrainMRI, and Br35H. This is attributed to the visual similarities between colon polyps and brain tumors. Conversely, the symptom of the colon polyp differs significantly from that of diseased skin or chest, leading to performance degradation in ISIC and COVID-19. Overall, compared to VAND, AnomalyCLIP performs consistently better across all datasets of anomaly detection and segmentation.

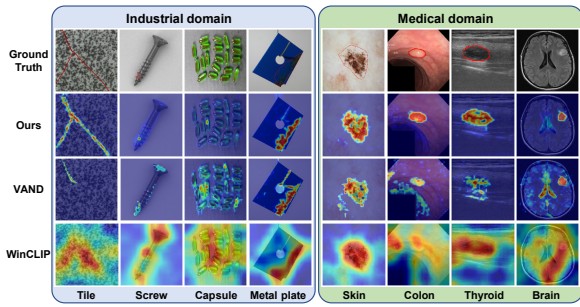

Figure 4: Segmentation visualization.

Table 3: ZSAD performance on medical iamges when fine-tuned by medical image datasets.

| Category | Datasets | VAND | AnomalyCLIP |
|---|---|---|---|
| Classification | | | |
| | HeadCT | (89.1, 89.4) | (93.5, 95.1) |
| Brain | BrainMRI | (89.3, 90.9) | (95.5, 97.2) |
| | Br35H | (93.1, 92.9) | (97.9, 98.0) |
| Chest | COVID-19 | (15.5, 8.5) | (70.9, 33.7) |
| Segmentation | | | |
| Skin | ISIC | (58.8, 31.2) | (83.0, 63.8) |
| | CVC-ClinicDB | (89.4, 82.3) | (92.4, 82.9) |
| Colon | Kvasir | (87.6, 39.3) | (92.5, 61.5) |
| | Endo | (88.5, 81.9) | (93.2, 84.8) |
| Thyroid | TN3K | (60.5, 16.8) | (79.2, 47.0) |

**Object-agnostic vs. object-aware prompt learning** To study the effectiveness of object-agnostic prompt learning in AnomalyCLIP, we compare AnomalyCLIP with its variant that uses an object-aware prompt template. The performance gain of AnomalyCLIP to its object-aware prompt learning variant is shown in Fig. 5, where positive values indicate our object-agnostic prompt templates are better than the object-aware one. It is clear that our object-agnostic

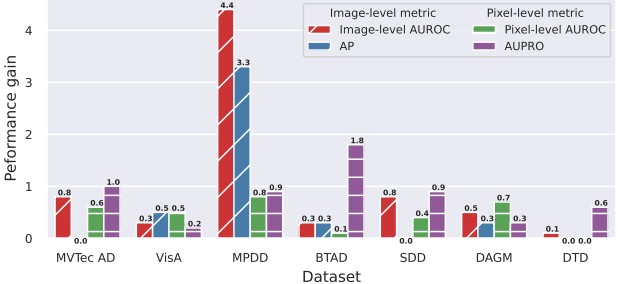

Figure 5: Performance gain of using object-agnostic prompts compared to object-aware prompts.

prompt learning performs much better than, or on par with, the object-aware version in both image-level and pixel-level anomaly detection. This indicates that having object-agnostic prompts helps better learn the generic abnormality and normality in images, as the object semantics are often not helpful, or can even become noisy features, for the ZSAD task.

## 4.3 ABLATION STUDY

**Module ablation** We first validate the effectiveness of different high-level modules of our AnomalyCLIP, including DPAM ($T_1$), object-agnostic text prompts ($T_2$), added learnable tokens in text encoders ($T_3$), and multi-layer visual encoder features ($T_4$). As shown in Table 4, each module contributes to the remarkable performance of AnomalyCLIP. DPAM improves the segmentation performance by enhancing local visual semantics ($T_1$). Object-agnostic text prompts focus on the abnormality/normality within images instead of the object semantics, allowing AnomalyCLIP to detect anomalies in diverse unseen objects. Therefore, introducing object-agnostic text prompts ($T_2$) significantly improves AnomalyCLIP. Furthermore, text prompt tuning ($T_3$) also brings performance improvement via the refinement of original textual space. Finally, $T_4$ integrates multi-layer visual semantics to provide more visual details, which further promotes the performance of ZSAD.

**Context optimization** Next we examine key modules in detail. The object-agnostic prompt learning is the most effective module, and it is driven by our glocal context optimization, so we consider two different optimization terms, local and global losses, in Eq. 2. The results are shown in Table 5. Both global and local context optimization contribute to the superiority of AnomalyCLIP. Global context optimization helps to capture global anomaly semantics, thus enabling more accurate image-level detection. Compared to global context optimization, local context optimization incorporates

Table 4: Module ablation.

| Module | MVTec AD | | VisA | |
|---|---|---|---|---|
| | Pixel-level | Image-level | Pixel-level | Image-level |
| Base | (46.8, 15.4) | (66.3, 83.3) | (47.9, 17.1) | (54.4, 61.7) |
| +$T_1$ | (68.4, 47.4) | (66.3, 83.3) | (54.8, 32.7) | (54.4, 61.7) |
| +$T_2$ | (89.5, 81.2) | (90.8, 96.0) | (95.0, 85.3) | (81.7, 85.2) |
| +$T_3$ | (90.0, 81.1) | (91.0, 96.1) | (95.2, 86.0) | (81.9, 85.2) |
| +$T_4$ | (91.1, 81.4) | (91.5, 96.2) | (95.5, 87.0) | (82.1, 85.4) |

Table 5: Context optimization ablation.

| Local. | Global. | MVTec AD | | VisA | |
|---|---|---|---|---|---|
| | | Pixel-level | Image-level | Pixel-level | Image-level |
| ✗ | ✗ | (71.7, 57.7) | (68.8, 85.8) | (74.7, 62.1) | (61.1, 69.1) |
| ✗ | ✓ | (80.3, 77.8) | (89.9, 95.4) | (86.6, 78.1) | (82.2, 84.9) |
| ✓ | ✗ | (91.0, 80.4) | (89.9, 96.0) | (95.2, 86.5) | (79.5, 83.2) |
| ✓ | ✓ | (91.1, 81.4) | (91.5, 96.2) | (95.5, 87.0) | (82.1, 85.4) |

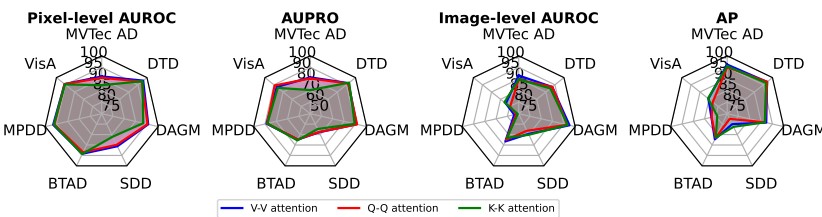

Figure 6: DPAM component ablation.

local anomaly semantics, which improves pixel-level performance and complements image-level performance. By synthesizing these two optimization strategies, AnomalyCLIP generally achieves better performance than using them individually.

**DPAM strategy ablation**   AnomalyCLIP uses $V$-$V$ self-attention by default. Here we study the effectiveness of using two other DPAM strategies, including $Q$-$Q$ and $K$-$K$ self-attention, resulting in two AnomalyCLIP variants, namely AnomalyCLIP$_{qq}$ and AnomalyCLIP$_{kk}$. The comparison results are presented in Fig. 6. AnomalyCLIP$_{qq}$ achieves similar segmentation capabilities as AnomalyCLIP but suffers from degradation in detecting image-level anomalies. Conversely, while AnomalyCLIP$_{kk}$ performs well in anomaly classification, its segmentation performance is less effective than AnomalyCLIP and AnomalyCLIP$_{qq}$. The $V$-$V$ self-attention is generally recommended in AnomalyCLIP. Detailed analysis of DPAM can be seen in Appendix C.

## 5   RELATED WORK

**Zero-shot anomaly detection**   ZSAD relies on the model's strong transferability to handle unseen anomalies (Aota et al., 2023). CLIP-AD (Liznerski et al., 2022) and ZOC (Esmaeilpour et al., 2022) are early studies in utilizing CLIP for ZSAD, but they mainly focus on the anomaly classification task. ACR (Li et al., 2023a) requires tuning on target-domain-relevant auxiliary data for ZSAD on different target datasets, while AnomalyCLIP can be applied to different datasets after it is trained on one general dataset. A very recent approach WinCLIP (Jeong et al., 2023) presents a seminal work that leverages CLIP for zero-shot classification and segmentation. It uses a large number of hand-crafted text prompts and involves multiple forward passes of image patches for anomaly segmentation. To tackle this inefficiency, VAND (Chen et al., 2023) introduces learnable linear projection techniques to enhance the modeling of local visual semantics. However, these approaches suffer from insufficiently generalized textual prompt embeddings, which degrades their performance in identifying anomalies associated with various unseen object semantics. AnomalyCLIP utilizes only two object-agnostic learnable text prompts to optimize the generic text prompts of abnormality and normality, and it can obtain segmentation results with just a single forward pass. AnomalyGPT (Gu et al., 2023) is a concurrent work in utilizing foundation models for AD, but it is designed for unsupervised/few-shot AD with manually crafted prompts.

**Prompt learning**   Rather than resorting to full network fine-tuning, prompt learning emerges as a parameter-efficient alternative to achieve satisfactory results (Sun et al., 2022; Zhou et al., 2022a; Khattak et al., 2023; Kim et al., 2023). CoOp (Zhou et al., 2022b) introduces learnable text prompts for few-shot classification. On this basis, DenseCLIP (Rao et al., 2022) extends prompt learning to dense prediction tasks with an extra image decoder. Instead, AnomalyCLIP proposes object-agnostic prompt learning for anomaly detection, blocking out the potential adverse impact of the diverse object semantics on anomaly detection. Benefiting from the glocal context optimization, AnomalyCLIP can capture local anomaly semantics such that we can simultaneously perform classification and segmentation tasks without an additional decoder network like Rao et al. (2022).

## 6   CONCLUSION

In this paper, we tackle a challenging yet significant area of anomaly detection, ZSAD, in which there is no available data in the target dataset for training. We propose AnomalyCLIP to improve the weak generalization performance of CLIP for ZSAD. We introduce object-agnostic prompt learning to learn generic abnormality/normality text prompts for generalized ZSAD on image datasets of diverse foreground objects. Further, to incorporate global and local anomaly semantics into AnomalyCLIP, we devise a joint global and local context optimization to optimize the object-agnostic text prompts. Extensive experimental results on 17 public datasets demonstrate that AnomalyCLIP achieves superior ZSAD performance.

REPRODUCIBILITY STATEMENT

To ensure the reproducibility and completeness of this paper, we have included an Appendix consisting of five main sections. In Appendix A, we provide more implementation details of AnomalyCLIP, as well as the reproduction of other baseline methods. Appendix B provides key statistics about the datasets used in our experiments and the implementation of the auxiliary medical dataset for prompt tuning. Appendix D supplements the main paper with additional results and ablations. Further visualizations of similarity scores and maps are detailed in Appendix E. Additionally, the main paper presents only the average performance in each dataset that contains a number of data subsets, for which we present their fine-grained detection results in Appendix F. Our code will be made publicly accessible once the paper is accepted.

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

## A  IMPLEMENTATION DETAILS AND BASELINES

### A.1  IMPLEMENTATION DETAILS

In this paper, we use the publicly available CLIP model (`VIT-L/14@336px`) as our backbone. Model parameters of CLIP are all frozen. The length of learnable text prompts $M$ is set to 12. These trainable text tokens are attached to the first 9 layers of the text encoder, and each text token has a length of 4. We fine-tune AnomalyCLIP on the test data on MVTec AD and test the performance for other datasets. As for MVTec AD, we fine-tune AomalyCLIP on test data on VisA. To provide adequate visual details, we extract local visual embeddings $v_m^i$ from the 6-th, 12-th, 18-th, and 24-th layers of the visual encoder. Starting from the 6-th layer, we apply DPAM to the architecture of the visual encoder according to Sec. 3.3. Additionally, we set the balanced weight $\lambda$ to 1 in our loss function. The input images are resized to a size of 518 with batch size 8, and we use the Adam optimizer (Kingma & Ba, 2014) with a learning rate of 0.001 to update model parameters. During testing, we apply a Gaussian filter with $\sigma = 4$ to smooth the anomaly score map. The epoch is 15 for all experiments, which are performed in PyTorch-2.0.0 with a single NVIDIA RTX 3090 24GB GPU.

### A.2  BASELINES

To demonstrate the superiority of Anomlay-CLIP, we compare AnomlayCLIP with broad SOTA baselines. Implementation and reproduction details are given as follows:

- CLIP (Radford et al., 2021). CLIP is a powerful zero-shot classification method. To perform the anomaly detection task, we use two classes of text prompt templates `A photo of a normal [cls]` and `A photo of an anomalous [cls]`, where `cls` denotes the target class name. The anomaly score is computed according to Eq. 1. As for anomaly segmentation, we extend the above computation to local visual embedding to derive the segmentation.

- CLIP-AC (Radford et al., 2021). Different from CLIP, CLIP-AC employs an ensemble of text prompt templates that are recommended for ImageNet dataset (Radford et al., 2021). We average the generated textual embeddings of normal and anomaly classes respectively, and compute the probability and segmentation in the same way as CLIP.

- WinCLIP (Jeong et al., 2023). WinCLIP is a SOTA ZSAD method. They design a large set of hand-crafted text prompt templates specific to anomaly detection and use a window scaling strategy to obtain anomaly segmentation. All parameters are kept the same as in their paper.

- VAND (Chen et al., 2023). VAND is an improved version of WinCLIP. They first adjust the text prompt templates and then introduce learnable linear projections to improve local visual semantics to derive more accurate segmentation. All parameters are kept the same as in their paper.

Table 6: Key statistics on the datasets used.

| Dataset | Category | Modalities | $\|\mathcal{C}\|$ | Normal and anomalous samples | Usage |
|---------|----------|------------|-----|------------------------------|-------|
| MVTec AD | Obj &texture | Photography | 15 | (467, 1258) | Industrial defect detection |
| VisA | | Photography | 12 | (962, 1200) | Industrial defect detection |
| MPDD | Obj | Photography | 6 | (176, 282) | Industrial defect detection |
| BTAD | | Photography | 3 | (451, 290) | Industrial defect detection |
| SDD | | Photography | 1 | (181, 74) | Industrial defect detection |
| DAGM | Texture | Photography | 10 | (6996, 1054) | Industrial defect detection |
| DTD-Synthetic | | Photography | 12 | (357, 947) | Industrial defect detection |
| ISIC | Skin | Photography | 1 | (0, 379) | Skin cancer detection |
| CVC-ClinicDB | | Endoscopy | 1 | (0, 612) | Colon polyp detection |
| CVC-ColonDB | | Endoscopy | 1 | (0, 380) | Colon polyp detection |
| Kvasir | | Endoscopy | 1 | (0, 1000) | Colon polyp detection |
| Endo | | Endoscopy | 1 | (0, 200) | Colon polyp detection |
| TN3K | Thyroid | Radiology (Utralsound) | 1 | (0, 614) | Thyroid nodule detection |
| HeadCT | | Radiology (CT) | 1 | (100, 100) | Brain tumor detection |
| BrainMRI | Brain | Radiology (MRI) | 1 | (98, 155) | Brain tumor detection |
| Br35H | | Radiology (MRI) | 1 | (1500, 1500) | Brain tumor detection |
| COVID-19 | Chest | Radiology (X-ray) | 1 | (1341, 219) | COVID-19 detection |

- CoOp (Zhou et al., 2022b). CoOp is a representative method for prompt learning. To adapt CoOp to ZSAD, we replace its learnable text prompt templates $[V_1][V_2]...[V_N][\texttt{cls}]$ with normality and abnormality text prompt templates, where $V_i$ is the learnable word embeddings. The normality text prompt template is defined as $[V_1][V_2]...[V_N][\texttt{normal}][\texttt{cls}]$, and the abnormality one is defined as $[V_1][V_2]...[V_N][\texttt{anomalous}][\texttt{cls}]$. Anomaly probabilities and segmentation are obtained in the same way as for AnomalyCLIP. All parameters are kept the same as in their paper.

## B  DATASET

**More dataset details**   In this paper, we conduct extensive experiments on 17 public datasets spanning two domains and three modalities to validate the effectiveness of our methods. Since we just use the test data of Datasets, we present the relevant information of their test sets in Table 6. We apply the default normalization of OpenCLIP to all datasets. After normalization, we resize the images to a resolution of (518, 518) to obtain an appropriate visual feature map resolution. It should be noted that the original image size of SDD has a width of 500 and a height ranging from 1,240 to 1,270. Before processing, we vertically divide the original 500 × 1,250 image into two images and assign pixel-wise annotations to each image.

**Fine-tuning medical dataset**   We cannot find publicly available 2D medical AD datasets that include both category labels and segmentation ground truths simultaneously. To fill the blank, in this paper, we create such a medical dataset by combining two existing 2D medical datasets. Particularly, we use the colon polyp detection dataset ColonDB (Tajbakhsh et al., 2015) to provide pixel-level annotations. Meanwhile, considering the normal samples in the same domain, we choose the test split of Endo classification dataset (Hicks et al., 2021) to combine with ColonDB. As a result, the new medical dataset contains 163 normal samples and 380 anomaly samples, supporting both anomaly classification and segmentation tasks.

## C   DETAILED ANALYSIS OF DPAM

Since the visual encoder of CLIP is originally pre-trained to align global object semantics, such as cat and dog, the contrastive loss used in CLIP makes the visual encoder produce a representative global embedding for recognizing semantic classes. Through the self-attention mechanism, the attention map in the visual encoder focuses on the specific tokens highlighted within the red rectangle in Fig. 3b. Although these tokens may contribute to global object recognition, they disrupt the local visual semantics, which directly hinders the effective learning of the fine-grained abnormality in our object-agnostic text prompts. For segmentation purposes, it's crucial for the visual feature map to emphasize the surrounding context to capture more local visual semantics.

Formally, let $a_{ij}$ be an attention score in the attention score matrix, where $i, j \in [1, h \times w]$, then the $i$-th output of $Q$-$K$ attention can be written as:

$$Attention(Q, K, V)_i = softmax\left(\frac{q_i K^\top}{\sqrt{D}}\right)V = \frac{\sum_{j=1}^{n} a_{ij} v_j}{\sum_{j=1}^{n} a_{ij}}, \qquad a_{ij} = e^{\frac{q_i k_j^\top}{\sqrt{D}}}.$$

Note that vectors (i.e., $q_i$, $k_i$, $v_i$) are represented as row vectors. $Attention(Q, K, V)_i$ can be regarded as the weighted average of $v_j$ using $a_{ij}$ as the weight. Assuming that the original attention map focuses on the specific tokens at index $m$, it is clear that $q_i$ only produces the large attention score with $k_m$ in all $k_j$. Therefore, $a_{im}$ is the largest score among other $a_{ij}$ so $Attention(Q, K, V)_i$ is dominated by $v_m$, which causes the local visual embedding at index $i$ to be disturbed by the local visual embedding at index $m$. In Figure 3(b), the attention score map presents vertical activation and suggests that every $q_i$ produces a large attention score with $k_m$. In such a case, several $Attention(Q, K, V)_i$ is dominated by $v_m$ and results in weak anomaly segmentation in Figure 3(b) even though $v_m$ may be important for original class recognition. Some prior studies (Rao et al., 2022; Gu et al., 2023) use an additional decoder to recover the local visual semantics. In this paper, we directly use local visual embeddings for segmentation and point out that an ideal attention map for local visual semantics should exhibit a more pronounced diagonal pattern. For this purpose, DPAM is proposed to replace the original $Q$-$K$ attention with analogous components, including $Q$-$Q$, $K$-$K$, and $V$-$V$ self-attention. Therefore, $a_{ij}$ is changed into:

$$a_{ij}^{qq} = e^{\frac{q_i q_j^\top}{\sqrt{D}}}, \quad a_{ij}^{kk} = e^{\frac{k_i k_j^\top}{\sqrt{D}}}, \quad a_{ij}^{vv} = e^{\frac{v_i v_j^\top}{\sqrt{D}}}.$$

This modification ensures that $q_i$, $k_i$, and $v_i$ hold significant weight in forming $Attention(Q, Q, V)_i$, $Attention(K, K, V)_i$, and $Attention(V, V, V)_i$, thereby preserving local visual semantics. As a result, the produced attention maps exhibit a more diagonal prominence compared to the original Q-K attention, leading to improved performance in anomaly segmentation, as shown in Fig.3c, Fig.3d, and Fig. 3e. However, since $Q$ and $K$ consist of the original attention map, other important tokens at index $n$ for class recognition within themselves may also produce relatively large scores ($a_{in}$) (e.g., $q_i$ has strong relevance with $q_n$ besides $q_i$) to disturb $Attention(Q, Q, V)_i$ and $Attention(K, K, V)_i$ Fig.3c and Fig.3d. In contrast to $Q$-$Q$ and $K$-$K$, $V$-$V$ does not participate in computing the original attention map, reducing the unexpected bias to different tokens in $V$ for the purpose of anomaly segmentation. Therefore, $v_i$ does not produce a large weight ($a_{ij}$) with $v_j$ and generates a larger weight ($a_{ii}$) to form $Attention(V, V, V)_i$, preserving more information of $v_i$ and experiencing diagonally prominent attention map (minimal disturbance), as depicted in Fig. 3e. This is the reason why $V$-$V$ achieves the best results.

## D   ADDITIONAL RESULTS AND ABLATIONS

**Module ablation by removing modules.**   We dive into the effectiveness of each module in AnomalyCLIP in Table 7. We test the contribution of one module by removing one module and maintaining the rest module.

1. The effectiveness of DPAM ($T_1$). When we remove DPAM, the results show a decrease from 91.1% AUROC to 87.9% AUROC in pixel-level performance and from 91.5% AUROC to 80.7% AUROC in image-level performance. This performance decline indicates

Table 7: Study on the effect of each module. $T_1$: DPAM, $T_2$: Object-agnostic prompt learning, $T_3$: Textual prompt tuning, and $T_4$: Integration of multi-scale local visual feature.

| T1 | T2 | T3 | T4 | MVTec AD | | VisA | |
|---|---|---|---|---|---|---|---|
| | | | | Pixel-level | Image-level | Pixel-level | Image-level |
| ✓ | ✓ | ✓ | ✓ | (91.1, 81.4) | (91.5, 96.2) | (95.5, 87.0) | (82.1, 85.4) |
| ✗ | ✓ | ✓ | ✓ | (87.9, 80.0) | (80.7, 89.5) | (91.9, 84.9) | (73.0, 77.7) |
| ✓ | ✗ | ✓ | ✓ | (84.3, 75.3) | (65.6, 85.3) | (89.5, 83.2) | (68.1, 72.9) |
| ✓ | ✓ | ✗ | ✓ | (90.6, 80.5) | (90.4, 95.3) | (94.5, 85.6) | (81.6, 85.1) |
| ✓ | ✓ | ✓ | ✗ | (90.0, 81.1) | (91.0, 96.1) | (95.2, 86.0) | (81.9, 85.2) |

Table 8: The study of computation overhead among baselines.

| Methods | Training time per epoch (min) | FPS | Performance (VisA) | |
|---|---|---|---|---|
| | | | Pixel-level | Image-level |
| CLIP | - | 13.23 | (46.6, 14.8) | (66.4, 71.5) |
| WinCLIP | - | 1.20 | (79.6, 56.8) | (78.1, 81.2) |
| VAND | 13.56 min | 9.23 | (94.2, 86.8) | (78.0, 81.4) |
| CoOp | 12.25 min | 12.75 | (24.2, 3.8) | (62.8, 68.1) |
| AnomalyCLIP | 13.71 min | 8.92 | (95.5, 87.0) | (82.1, 85.4) |
| AnomalyCLIP without DPAM | 12.98 min | 10.21 | (91.9, 84.9) | (73.0, 77.7) |

the importance of DPAM, which enhances local visual semantics by modifying the attention mechanism. However, the decrease in performance at the image level is more pronounced than that at the pixel level. This discrepancy is attributed to the fact that the total loss places greater emphasis on local context optimization, driven by a larger local loss compared to the case with DPAM.

2. The effectiveness of object-agnostic prompt learning ($T_2$). Excluding object-agnostic prompt learning makes AnomalyCLIP suffer from the huge performance gap (i.e., 91.1% AUROC to 84.3% AUROC in pixel-level and 91.5% AUROC to 65.6% AUROC in image-level). This performance decline illustrates that the object-agnostic text prompt template plays a significant role in improving the performance of AnomalyCLIP at both pixel and image levels.

3. The effectiveness of textual prompt tuning ($T_3$). When removing textual prompt tuning, the performance of AnomalyCLIP declines from 91.1% AUROC to 90.6% AUROC in pixel-level performance and from 91.5% AUROC to 90.4% AUROC in image-level performance. This demonstrates the importance of adapting original textual space by adding learnable textual tokens in the text encoder.

4. The effectiveness of the integration of multi-layer local visual semantics ($T_4$). When removing multi-layer local visual semantics, the outcomes reveal a decrease from 91.1% AUROC to 90.0% AUROC in pixel-level performance and from 91.5% AUROC to 91.0% AUROC in image-level performance. This performance decline indicates the importance of incorporating multi-layer local visual semantics.

**Study of computation overhead** In addition to performance, computation overhead is also an important metric to evaluate the model. Therefore, we assess the time taken during training (training time per epoch) and the inference speed (frames per second, FPS). For a fair comparison, all experiments are conducted in a single 3090 NVIDIA RTX 3090 24GB GPU, and the GPU is kept free before evaluation. In Table 8, AnomalyCLIP takes 13.71 min per epoch on MVTec AD (The total number of samples is 1725) and only requires a total of 15 epochs for the whole fine-tuning. Once AnomalyCLIP finishes fine-tuning, AnomalyCLIP can be applied to different datasets and domains without additional training. We also compare AnomalyCLIP with other baselines that need auxiliary data (i.e., CoOp and VAND). The minimum training time per epoch is 12.25 min of CoOp, and hence the training time taken is similar for fine-tuning methods. As for inference speed, CLIP achieves the 13.23 FPS. However, it suffers from weak detection performance. Although WinCLIP achieves better performance, WinCLIP has only 1.2 FPS because it needs multiple forward image patches to derive the segmentation. AnomalyCLIP outperforms WinCLIP and obtains 8.92 FPS.

We also evaluated the computation overhead of DPAM separately. In Table 8, without DPAM, AnomalyCLIP takes 12.98 min to train per epoch. Compared to the 13.71 min for AnomalyCLIP with DPAM, we observe that introducing DPAM does not significantly increase the time complexity.

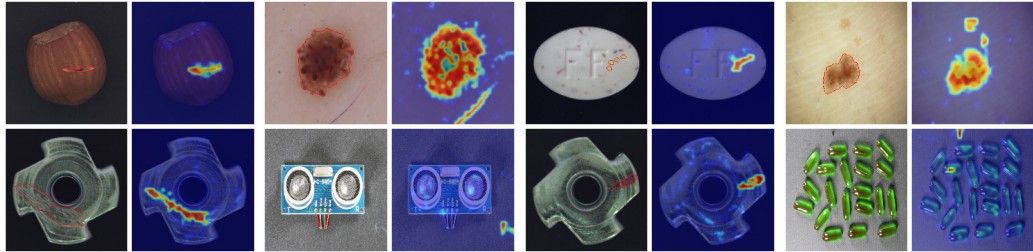

(a) Successful case of de-(b) Failure case of detect-(c) Successful case of de-(d) Failure case of detect-
tecting scratch-like defects.ing scratch-like defects.    tecting color stain.          ing color stain

Figure 7: Analysis for successful and failure cases.

Table 9: Study on the effect of shared and unshared learnable word embeddings.

|  | MVTec AD | | VisA | |
| --- | --- | --- | --- | --- |
|  | Pixel-level | Image-level | Pixel-level | Image-level |
| Shared | (90.5, 80.1) | (90.9, 95.2) | (95.0, 86.4) | (81.5, 84.4) |
| Unshared | (91.1, 81.4) | (91.5, 96.2) | (95.5, 87.0) | (82.1, 85.4) |

This is attributed to the fact that DPAM only creates the two paths during the computation of the attention map and is frozen during fine-tuning, thereby avoiding the computationally expensive process of gradient computation. Meanwhile, DPAM also does not result in large computation overhead during inference: AnomalyCLIP w/ DPAM gets 8.92 FPS vs. 10.21 FPS for w/o using DPAM.

**Discussion of successful and failure case**    Although AnomalyCLIP achieves superior results in ZSAD, we find that there still exist some potential failures. In Fig. 7a, AnomalyCLIP accurately detects scratch-like patterns on the product, even when they typically appear in the texture. However, false detection occurs when scratch-like patterns are situated in the background, as depicted in Fig. 7b. Meanwhile, we also show the color stain pattern. As shown in Fig. 7c, AnomalyCLIP successfully detects the color stain, which exhibits subtle visual differences from the detected entities. However, AnomalyCLIP may face challenges when the normal region displays patterns that are indistinguishable to the naked eye from anomalies. For instance, in skin cancer detection, the normal regions falsely detected as anomalies are actually visually similar to the disease region in Fig. 7c. Also, the stain interference in the background is also a problem. These failure cases illustrate the importance of mitigating background interference and achieving fine-grained discrimination, especially in cases of visually similar abnormalities. Exploring these challenges for enhancing ZSAD is a valuable direction for future research.

**Study on the effect of shared and unshared learnable word embeddings**    As presented in Table 9, when sharing the learnable word embeddings of $g_n$ and $g_a$, AnomalyCLIP achieves 90.5% AUROC in pixel level and 90.9% in image-level on MVTec AD and 95.0% AUROC in pixel level and 81.5% AUROC in image level on VisA. The results show that AnomalyCLIP without sharing also works well for ZSAD and the efficiency of our object-agnostic prompt learning. However, the shared prompt performs slightly worse than the unshared prompts (used in the original paper). The performance decrease is 0.6%AUROC and 0.6%AUROC in image level on MVTec AD and Visa, and 0.5%AUROC and 0.6%AUROC in pixel level. We believe that the separate learning for these two prompts helps discriminate the generic normality and abnormality because when we share the parameters of $V_i$ and $W_i$, the learned semantics of normal and anomaly may be confused.

**Study on the effect of local visual features**    Here, we examine the impact of various types of local visual semantics. We explore two ensemble methods, namely AnomalyCLIP$_{ensemble1}$ and AnomalyCLIP$_{ensemble2}$, involving the ensemble of $Q$-$Q$, $K$-$K$, and $V$-$V$ and the ensemble of $Q$-$K$, $Q$-$Q$, $K$-$K$, and $V$-$V$, respectively. In addition to $Q$-$K$ and $V$-$V$ features, we average the logit output of different features for the ensemble. As shown in Table 10, AnomalyCLIP$_{ensemble1}$ shows performance improvement by leveraging the advantages of three DPAM features. However, while AnomalyCLIP$_{ensemble2}$ outperforms the $Q$-$K$ feature version, it experiences a perfor-

Table 10: Study on the effect of local visual features.

| Module | MVTec AD | | VisA | |
|---|---|---|---|---|
| | Pixel-level | Image-level | Pixel-level | Image-level |
| Q-K | (87.9, 80.0) | (80.7, 89.5) | (91.9, 84.9) | (73.0, 77.7) |
| V-V | (91.1, 81.4) | (91.5, 96.2) | (95.5, 87.0) | (82.1, 85.4 ) |
| Q-Q, K-K, V-V (ensemble) | (91.2, 83.3) | (91.4, 96.1) | (95.8, 87.6) | (82.3, 85.7) |
| Q-K, Q-Q, K-K, V-V(ensemble) | (90.7, 80.5) | (90.8, 96.0) | (94.9, 86.6) | (80.7, 83.7) |

mance decrease compared to $V$-$V$ from 91.1%AUROC to 90.7%AUROC and 91.5%AUROC to 90.8%AUROC on MVTec AD. There is also a decline from 95.5%AUROC to 94.9%AUROC and 82.1%AUROC to 80.7%AUROC on VisA. The decline in performance upon adding $Q$-$K$ features to AnomalyCLIP$_{ensemble1}$ suggests that the $Q$-$K$ feature fails to provide valid local visual semantics to facilitate ZSAD. Note that the original CLIP exploits $Q$-$K$ features and gets the weak segmentation performance. The seemingly good pixel-level performance of $Q$-$K$ in AnomalyCLIP is attributed to local optimization, where the object-agnostic prompt helps alleviate the disrupted local visual semantics of $Q$-$K$.

Table 11: Ablation on the effect of the Focal and Dice loss.

| Focal loss | Dice loss | MVTec AD | | VisA | |
|---|---|---|---|---|---|
| | | Pixel-level | Image-level | Pixel-level | Image-level |
| ✗ | ✗ | (80.3, 77.8) | (89.9, 95.4) | (86.6, 78.1) | (82.2, 84.9) |
| ✗ | ✓ | (87.2, 78.6) | (89.5, 95.2) | (90.1, 79.8) | (81.8, 85.2) |
| ✓ | ✗ | (90.6, 78.1) | (91.0, 96.0) | (94.9, 86.1) | (81.2, 84.6) |
| ✓ | ✓ | (91.1, 81.4) | (91.5, 96.2) | (95.5, 87.0) | (82.1, 85.4) |

**Focal and Dice loss ablation** Focal and Dice loss play a crucial role in optimizing local context. They are introduced to empower object-agnostic text prompts to focus on fine-grained, local abnormal regions from intermediate layers of the visual encoder. As mentioned in Section 3.2, Focal loss addresses the imbalance between anomaly and normal pixels, typically caused by the smaller size of anomalous regions. Meanwhile, Dice loss aims to precisely constrain the anomaly boundary by measuring the overlap between the predicted segmentation ($S_n/S_a$) and the ground truth mask. To provide a more comprehensive analysis, we have included an ablation study on Focal and Dice loss in Table 11. Compared to scenarios without local context optimization, Dice loss improves the pixel-level and image-level performance from 80.3%AUROC to 87.2%AUROC and 86.6%AUROC to 90.1%AUROC in pixel level on MVTec AD and VisA. Focal loss also brings the performance gain of 10.3%AUROC and 8.3%AUROC. Combining Focal and Dice loss, AnomalyCLIP achieves the best results (i.e., 91.1%AUROC and 95.5%AUROC). Note that the global context optimization is always used during the ablation, since we need at least one loss function to drive the optimization.

Table 12: Comparison of ZSAD performance between AnomalyCLIP and SOTA full-shot methods. The best performance is highlighted in red, and the second-best is highlighted in blue.

| Task | Category | Datasets | $|\mathcal{C}|$ | AnomalyCLIP | PatchCore | RD4AD |
|---|---|---|---|---|---|---|
| Image-level (AUROC, AP) | Obj &texture | MVTec AD | 15 | (91.5, 96.2) | (99.0, 99.7) | (98.7, 99.4) |
| | | VisA | 12 | (82.1, 85.4) | (94.6, 95.9) | (95.3, 95.7) |
| | Obj | MPDD | 6 | (77.0, 82.0) | (94.1, 96.3) | (91.6, 93.8) |
| | | BTAD | 3 | (88.3, 87.3) | (93.2, 98.6) | (93.8, 96.8) |
| | | SDD | 1 | (84.7, 80.0) | (64.9, 48.3) | (86.8, 81.3) |
| | Texture | DAGM | 10 | (97.5, 92.3) | (92.7, 81.3) | (92.9, 79.1) |
| Pixel-level (AUROC, PRO) | Obj &texture | MVTec AD | 15 | (91.1, 81.4) | (98.1, 92.8) | (97.8, 93.6) |
| | | VisA | 12 | (95.5, 87.0) | (98.5, 92.2) | (98.4, 91.2) |
| | Obj | MPDD | 6 | (96.5, 88.7) | (98.8, 94.9) | (98.4, 95.2) |
| | | BTAD | 3 | (94.2, 74.8) | (97.4, 74.4) | (97.5, 75.1) |
| | | SDD | 1 | (90.6, 67.8) | (87.9, 46.3) | (92.2, 72.0) |
| | Texture | DAGM | 10 | (95.6, 91.0) | (95.9, 87.9) | (96.8, 91.9) |

**Comparison with SOTA full-shot methods** In this section, we are interested in the performance gap between AnomalyCLIP and the recently published SOTA full-shot methods, such as Patch-Core (Roth et al., 2022) and RD4AD (Deng & Li, 2022). Since some datasets do not provide normal training data, we conduct experiments on six public datasets. As shown in Table 12, AnomalyCLIP achieves comparable anomaly detection and segmentation performance compared to PatchCore and RD4AD, and it even outperforms them in some datasets. This illustrates that the generic prompt

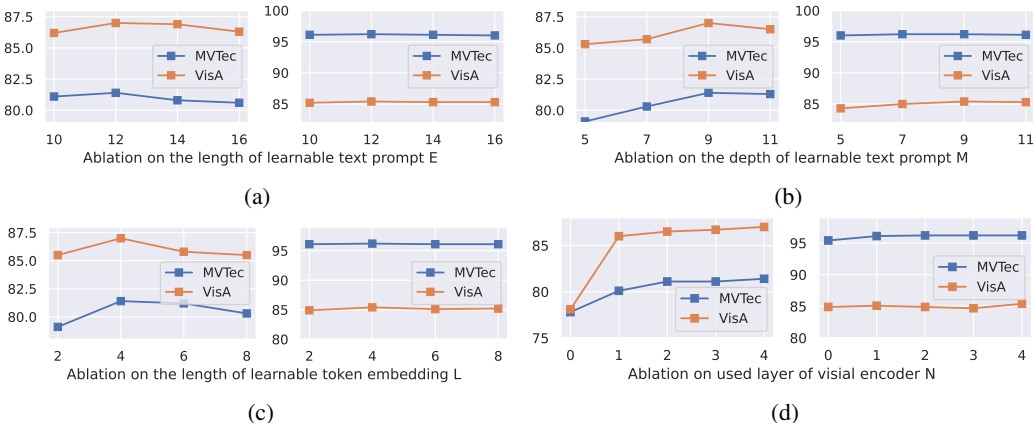

Figure 8: Hyparameter analysis. (a)$E$ ablation. (b) $M$ ablation. (c)$L$ ablation (d)$N$ ablation. Pixel/image-level (AUPRO, AP) performances are shown on the left and right sides of each subplot, respectively.

embeddings empower AnomalyCLIP to effectively capture the normality and abnormality so that AnomalyCLIP can surpass the performance boundary decided by the training data.

**Refinement of the textual space** A representative embedding is not only decided by the well-designed text prompt, it also depends on the appropriate textual space. During fine-tuning, randomly initialized learnable token embeddings are introduced in the text encoder to refine the textual space for the adaption to AD. To control the degree of refining the textual space, we choose to insert the learnable token embeddings into the text encoder from its bottom to the top layer. In particular, the trainable and original tokens are denoted as $t'_m$ and $t_m$, respectively, where $m$ represents the layer of the text encoder. To integrate the original textual representations, for the layer $m$, we concatenate $t'_m$ and $t_m$ along the dimension of the channel and then forward them into $T_m$ to get $r'_{m+1}$ and $t_{m+1}$. Due to the self-attention mechanism, the output of $t_{m+1}$ contains the information of $t'_m$. In order to provide adequate calibration, we discard the obtained $r'_{m+1}$ and initialize new learnable token embeddings $t'_{m+1}$. Through this operation, $t'_{m+1}$ further refines textual representations of the layer $m + 1$. We repeat this operation until we reach the designated layer $M'$. This procedure is given by:

$$[r'_{m+1}, t_{m+1}] = T_m([t'_m, t_m])$$
$$[r'_{m+2}, t_{m+2}] = T_{m+1}([t'_{m+1}, t_{m+1}]) \tag{3}$$
$$\dots$$
$$t_{M'+1} = T_{M'}(t_{M'}),$$

where the operator $[\cdot, \cdot]$ represents the concatenation along the channel.

**Hyparameter analysis** We study the length of learnable text prompts $E$, depth of learnable token embeddings $M$, length of learnable token embeddings $M$, and number of used layers in visual encoder $N$. As shown in Fig. 8b, we observe that the detection and segmentation performance initially improves with an increase in the value of $E$. However, within the range of lengths from 12 to 16, we notice a decline in performance, which suggests that excessively long learnable text prompts could involve redundant information. Therefore, an appropriate value for $E$, such as $E = 12$, is beneficial to accurate learning of object-agnostic text prompts. Besides, we also investigate the depth of the attached learnable token embeddings in Fig. 8b. The degree of refining of the initial text space becomes more pronounced as the depth increases, enabling more discriminative textual embeddings for normal and anomaly. However, the performance drops when the refinement is excessive and impairs the generalization of AnomlayCLIP, as seen in the case when $M$ equals 9. After selecting the depth, we proceed to investigate the influence of the length of learnable token embeddings. As illustrated in Fig. 8c, we find that the length of token embeddings also involves a similar tradeoff between the model generalization and calibration of textual space in Fig. 8d. AnomalyCLIP achieves the overall performance gain when we provide the most local visual semantics ($N = 4$).

Table 13: Ablation on the robustness of the abnormality-related token in our prompt template on industrial defect datasets.

| Task | Category | Datasets | damaged | anomalous | flawed | defective | blemished |
|---|---|---|---|---|---|---|---|
| Image-level (AUROC, AP) | Obj &texture | MVTec AD | (91.5, 96.2) | (91.4, 96.2) | (91.3, 96.2) | (91.4, 96.2) | (91.5, 96.2) |
| | | VisA | (82.1, 85.4) | (80.7, 84.5) | (80.7, 84.5) | (80.9, 84.6) | (80.7, 84.5) |
| | Obj | MPDD | (77.0, 82.0) | (78.0, 83.9) | (77.9, 83.6) | (77.8, 83.5) | (78.6. 84.1) |
| | | BTAD | (88.3, 87.3) | (84.8, 86.7) | (85.2, 87.4) | (84.8, 86.2) | (85.9, 67.1) |
| | | SDD | (84.7, 80.0) | (82.3, 76.3) | (82.6, 76.8) | (82.8, 77.2) | (82.7, 77.0) |
| | Texture | DAGM | (97.5, 92.3) | (97.7, 92.6) | (97.5, 92.4) | (97.5, 92.3) | (97.5, 92.4) |
| | | DTD-Synthetic | (93.5, 97.0) | (93.3, 96.9) | (93.2, 96.9) | (93.4, 97.0) | (93.5, 97.0) |
| Pixel-level (AUROC, PRO) | Obj &texture | MVTec AD | (91.1, 81.4) | (91.0, 81.4) | (90.7, 81.4) | (91.0, 81.7) | (90.9, 81.2) |
| | | VisA | (95.5, 87.0) | (95.5, 86.5) | (95.5, 86.5) | (95.5, 86.2) | (95.6, 86.5) |
| | Obj | MPDD | (96.5, 88.7) | (96.6, 88.4) | (96.7, 89.0) | (96.7, 89.2) | (96.6, 88.8) |
| | | BTAD | (94.2, 74.8) | (94.3, 74.3) | (94.4, 75.1) | (94.3, 75.2) | (94.3, 73.7) |
| | | SDD | (90.6, 67.8) | (89.6, 66.8) | (89.5, 66.5) | (89.5, 64.8) | (89.6, 64.6) |
| | Texture | DAGM | (95.6, 91.0) | (95.6, 91.2) | (95.6, 91.3) | (95.5, 90.9) | (95.6, 90.9) |
| | | DTD-Synthetic | (97.9, 92.3) | (97.9, 92.3) | (97.9, 92.1) | (97.9, 92.5) | (97.9, 92.2) |

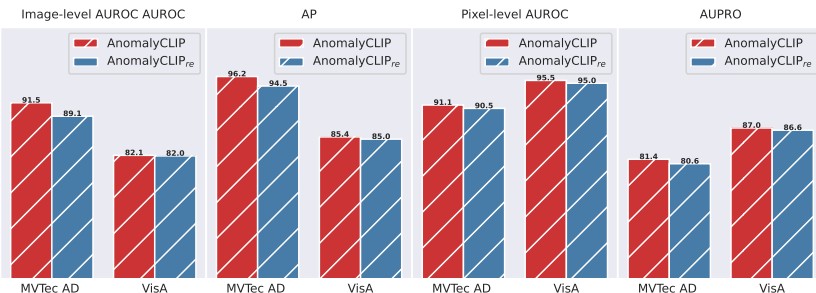

Figure 9: Object ablation.

**Prompt template ablation** Here, we study the robustness of AnomalyCLIP to prior anomaly semantics in the object-agnostic text prompt template. We replace `damaged` in the object-agnostic text prompt with other words having similar anomaly semantics, such as `anomalous`, `flawed`, `defective`, `blemished`. The results are presented in Table 13 and Table 14. The steady results indicate that AnomalyCLIP is not sensitive to the prior anomaly semantics introduced by the object-agnostic text prompt template.

**Object ablation** To investigate what the object-agnostic text prompts have learned, we replace `object` in object-agnostic text prompts with specific target `[cls]`, resulting in AnomalyCLIP$_{re}$. In Fig. 9, AnomalyCLIP$_{re}$ still performs well in ZSAD, even as we block out the object semantics during fine-tuning. This suggests that the knowledge learned by object-agnostic text prompts is the underlying anomaly patterns, allowing them to provide discriminative textual embeddings even when specific object semantics are incorporated. Furthermore, compared to AnomalyCLIP, AnomalyCLIP$_{re}$ shows a performance decay, which can be attributed to the inclusion of redundant/noisy object semantics. These results once again demonstrate the generalization ability of object-agnostic prompt learning.

Table 14: Ablation on the robustness of the abnormality-related token in our prompt template on medical image datasets.

| Task | Category | Datasets | damaged | anomalous | flawed | defective | blemished |
|---|---|---|---|---|---|---|---|
| Image-level (AUROC, AP) | Brain | HeadCT | (93.4, 91.6) | (93.1, 90.6) | (93.3, 90.8) | (93.5, 91.0) | (93.8, 91.5) |
| | | BrainMRI | (90.3, 92.2) | (87.8, 90.4) | (87.7, 90.0) | (88.3, 90.5) | (88.6, 90.7) |
| | | Br35H | (94.6, 94.7) | (93.1, 93.0) | (92.9, 92.8) | (93.1, 93.0) | (93.2, 93.1) |
| | Chest | COVID-19 | (80.1, 58.7) | (80.0, 58.5) | (80.2, 58.8) | (80.6, 59.0) | (82.1, 61.4) |
| Pixel-level (AUROC, PRO) | Skin | ISIC | (89.7, 78.4) | (90.1, 80.1) | (90.1, 80.1) | (90.4, 81.0) | (90.2, 80.6) |
| | Colon | CVC-ColonDB | (81.9, 71.3) | (82.2, 71.5) | (82.3, 71.6) | (82.1, 71.1) | (82.2, 71.5) |
| | | CVC-ClinicDB | (82.9, 67.8) | (83.0, 68.1) | (83.1, 68.4) | (82.9, 67.9) | (83.1, 68.2) |
| | | Kvasir | (78.9, 45.6) | (79.4, 45.1) | (79.4, 45.2) | (79.3, 44.9) | (79.5, 45.8) |
| | | Endo | (84.1, 63.6) | (84.3, 63.5) | (84.2, 63.5) | (84.2, 62.9) | (84.3, 63.4) |
| | Thyroid | TN3K | (81.5, 50.4) | (81.5, 51.7) | (81.3, 50.9) | (81.3, 50.3) | (81.6, 51.1) |

# E    VISUALIZATION

**Similarity score between textual and visual embeddings.**    We present visualizations of the similarity scores generated by both CLIP and AnomalyCLIP. These visualizations aim to provide an intuitive illustration of the effective adaptation made by AnomalyCLIP in comparison to CLIP. As shown in Fig. 10 and Fig. 11, we present the similarity score of CLIP on MVTec AD and VisA. The normal and anomaly scores are severely overlapped. Further, the range of scores is centered at 0.5. These show that the textual and visual space of CLIP originally aligned for object semantics are not desired for ZSAD. Also, we visualize the similarity scores of AnomalyCLIP in Fig. 12 and Fig. 13. Compared to CLIP, there is a significant overlap between the scores assigned to normal and anomaly instances, and at the same time, the score range is considerably wider. These results indicate that AnomalyCLIP achieves a significant improvement in adapting CLIP to ZSAD.

**Anomaly score map for different datasets.**    In addition to the similarity score for anomaly classification, we also visualize the anomaly score maps to present the strong anomaly segmentation ability of AnomalyCLIP. Specifically, we visualize the industrial object class: hazelnut, pill, and screw from MVTec AD; candle, chewinggum, capsule, cashew, pcb, and pip fryum from Visa; bracket, metal plate, and tube from MPDD. We also visualize the industrial texture: grid, leather, carpet, tile, wood, and zipper. In addition, we visualize the segmentation in medical domain across photography, endoscopy, and radiology images: skin cancer detection from ISIC; thyroid nodule detection from TN3K; colon polyp detection from Kvasir; brain tumor detection from Br35H.

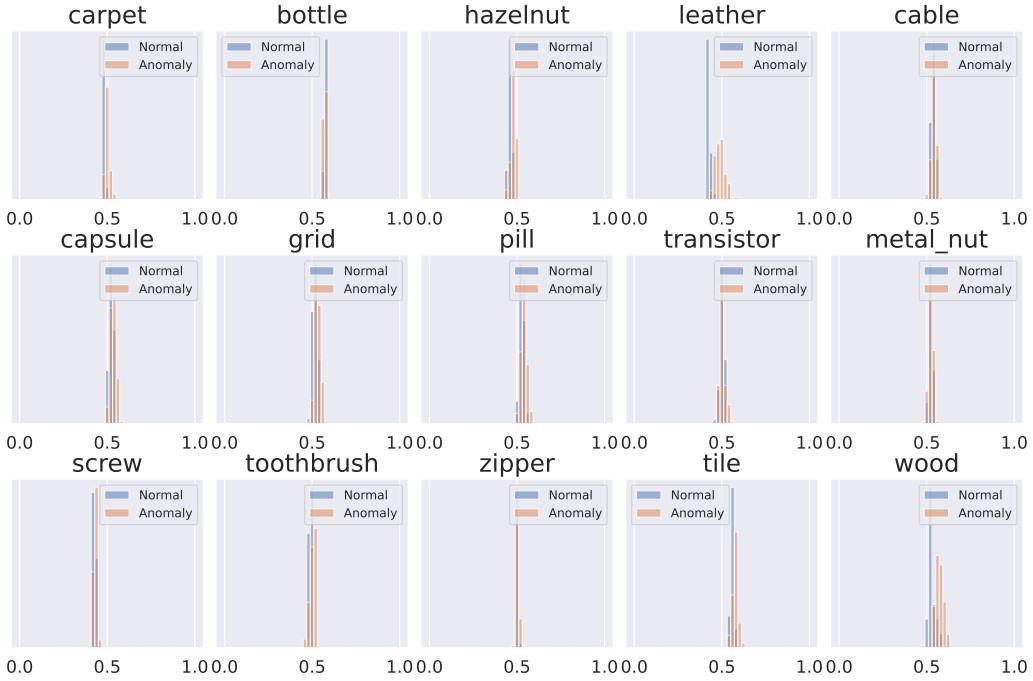

Figure 10: Similarity scores of CLIP on MVTec AD. Each sub-figure represents the visualization of one object.

# F    FINE-GRAINED ZSAD PERFORMANCE

In this section, we present the fine-grained data subset-level ZSAD performance in details.

Table 15: Fine-grained data-subset-wise performance comparison (AUROC) for anomaly segmentation on MVTec AD.

| Object name | CLIP | CLIP-AC | WinCLIP | VAND | CoOp | AnomalyCLIP |
|---|---|---|---|---|---|---|
| Carpet | 11.5 | 10.7 | 95.4 | 98.4 | 6.7 | 98.8 |
| Bottle | 17.5 | 23.3 | 89.5 | 83.4 | 23.1 | 90.4 |
| Hazelnut | 25.2 | 34.0 | 94.3 | 96.1 | 30.2 | 97.1 |
| Leather | 9.9 | 5.6 | 96.7 | 99.1 | 11.7 | 98.6 |
| Cable | 37.4 | 37.5 | 77.0 | 72.3 | 49.7 | 78.9 |
| Capsule | 50.9 | 49.1 | 86.9 | 92.0 | 35.5 | 95.8 |
| Grid | 8.7 | 11.9 | 82.2 | 95.8 | 7.8 | 97.3 |
| Pill | 55.8 | 60.8 | 80.0 | 76.2 | 46.5 | 92 |
| Transistor | 51.1 | 48.5 | 74.7 | 62.4 | 50.1 | 71 |
| Metal_nut | 43.9 | 53.6 | 61.0 | 65.4 | 49.3 | 74.4 |
| Screw | 80.1 | 76.4 | 89.6 | 97.8 | 17.0 | 97.5 |
| Toothbrush | 36.3 | 35.0 | 86.9 | 95.8 | 64.9 | 91.9 |
| Zipper | 51.5 | 44.7 | 91.6 | 91.1 | 33.4 | 91.4 |
| Tile | 49.9 | 39.1 | 77.6 | 92.7 | 41.7 | 94.6 |
| Wood | 45.7 | 42.4 | 93.4 | 95.8 | 31.4 | 96.5 |
| Mean | 38.4 | 38.2 | 85.1 | 87.6 | 33.3 | 91.1 |

Table 16: Fine-grained data-subset-wise performance comparison (PRO) for anomaly segmentation on MVTec AD.

| Object name | CLIP | CLIP-AC | WinCLIP | VAND | CoOp | AnomalyCLIP |
|---|---|---|---|---|---|---|
| Carpet | 2.9 | 1.9 | 84.1 | 48.5 | 0.5 | 90.1 |
| Bottle | 1.4 | 4.9 | 76.4 | 45.6 | 4.5 | 80.9 |
| Hazelnut | 2.8 | 9.4 | 81.6 | 70.3 | 4.7 | 92.4 |
| Leather | 0.2 | 0.0 | 91.1 | 72.4 | 1.8 | 92.2 |
| Cable | 7.3 | 6.9 | 42.9 | 25.7 | 12.2 | 64.4 |
| Capsule | 13.2 | 14.9 | 62.1 | 51.3 | 5.7 | 87.2 |
| Grid | 0.9 | 2.4 | 57.0 | 31.6 | 1.0 | 75.6 |
| Pill | 6.0 | 8.2 | 65.0 | 65.4 | 3.2 | 88.2 |
| Transistor | 15.3 | 11.2 | 43.4 | 21.3 | 9.3 | 58.1 |
| Metal_nut | 2.9 | 10.3 | 31.8 | 38.4 | 7.0 | 71.0 |
| Screw | 57.8 | 56.2 | 68.5 | 67.1 | 6.4 | 88.0 |
| Toothbrush | 5.8 | 5.2 | 67.7 | 54.5 | 16.6 | 88.5 |
| Zipper | 17.7 | 15.2 | 71.7 | 10.7 | 11.6 | 65.3 |
| Tile | 21.5 | 16.3 | 51.2 | 26.7 | 10.1 | 87.6 |
| Wood | 13.7 | 10.3 | 74.1 | 31.1 | 5.1 | 91.2 |
| Mean | 11.3 | 11.6 | 64.6 | 44.0 | 6.7 | 81.4 |

Table 17: Fine-grained data-subset-wise performance comparison (AUROC) for anomaly classification on MVTec AD.

| Object name | CLIP | CLIP-AC | WinCLIP | VAND | CoOp | AnomalyCLIP |
|---|---|---|---|---|---|---|
| Carpet | 96 | 93.1 | 100.0 | 99.5 | 99.9 | 100.0 |
| Bottle | 45.9 | 46.1 | 99.2 | 92.0 | 87.7 | 89.3 |
| Hazelnut | 88.7 | 91.1 | 93.9 | 89.6 | 93.5 | 97.2 |
| Leather | 99.4 | 99.5 | 100.0 | 99.7 | 99.9 | 99.8 |
| Cable | 58.1 | 46.6 | 86.5 | 88.4 | 56.7 | 69.8 |
| Capsule | 71.4 | 68.8 | 72.9 | 79.9 | 81.1 | 89.9 |
| Grid | 72.5 | 63.7 | 98.8 | 86.3 | 94.7 | 97.0 |
| Pill | 73.6 | 73.8 | 79.1 | 80.5 | 78.6 | 81.8 |
| Transistor | 48.8 | 51.2 | 88.0 | 80.8 | 92.2 | 92.8 |
| Metal_nut | 62.8 | 63.4 | 97.1 | 68.4 | 85.3 | 93.6 |
| Screw | 78.2 | 66.7 | 83.3 | 84.9 | 88.9 | 81.1 |
| Toothbrush | 73.3 | 89.2 | 88.0 | 53.8 | 77.5 | 84.7 |
| Zipper | 60.1 | 36.1 | 91.5 | 89.6 | 98.8 | 98.5 |
| Tile | 88.5 | 89.0 | 100.0 | 99.9 | 99.7 | 100.0 |
| Wood | 94 | 94.9 | 99.4 | 99.0 | 97.7 | 96.8 |
| Mean | 74.1 | 71.5 | 91.8 | 86.1 | 88.8 | 91.5 |

Table 18: Fine-grained data-subset-wise performance comparison (AP) on for anomaly classification MVTec AD.

| Object name | CLIP | CLIP-AC | WinCLIP | VAND | CoOp | AnomalyCLIP |
|---|---|---|---|---|---|---|
| Carpet | 98.8 | 97.8 | 100.0 | 99.8 | 100.0 | 100.0 |
| Bottle | 78.9 | 79.8 | 99.8 | 97.7 | 96.4 | 97.0 |
| Hazelnut | 94.6 | 95.9 | 96.9 | 94.8 | 96.7 | 98.6 |
| Leather | 99.8 | 99.8 | 100.0 | 99.9 | 100.0 | 99.9 |
| Cable | 70.8 | 64.3 | 91.2 | 93.1 | 69.4 | 81.4 |
| Capsule | 92.1 | 90.9 | 91.5 | 95.5 | 95.7 | 97.9 |
| Grid | 87.1 | 83.9 | 99.6 | 94.9 | 98.1 | 99.1 |
| Pill | 93.4 | 93.6 | 95.7 | 96.0 | 94.2 | 95.4 |
| Transistor | 48.1 | 49.9 | 87.1 | 77.5 | 90.2 | 90.6 |
| Metal_nut | 87.7 | 89.2 | 99.3 | 91.9 | 96.3 | 98.5 |
| Screw | 91.4 | 86.6 | 93.1 | 93.6 | 96.2 | 92.5 |
| Toothbrush | 90.7 | 96.0 | 95.6 | 71.5 | 90.4 | 93.7 |
| Zipper | 87.4 | 73.9 | 97.5 | 97.1 | 99.7 | 99.6 |
| Tile | 95.9 | 96.2 | 100.0 | 100.0 | 99.9 | 100.0 |
| Wood | 97.9 | 98.3 | 99.8 | 99.7 | 99.4 | 99.2 |
| Mean | 87.6 | 86.4 | 96.5 | 93.5 | 94.8 | 96.2 |

Table 19: Fine-grained data-subset-wise performance comparison (AUROC) for anomaly segmentation on VisA.

| Object name | CLIP | CLIP-AC | WinCLIP | VAND | CoOp | AnomalyCLIP |
|---|---|---|---|---|---|---|
| Candle | 33.6 | 50.0 | 88.9 | 97.8 | 16.3 | 98.8 |
| Capsules | 56.8 | 61.5 | 81.6 | 97.5 | 47.5 | 95.0 |
| Cashew | 64.5 | 62.5 | 84.7 | 86.0 | 32.5 | 93.8 |
| Chewinggum | 43.0 | 56.5 | 93.3 | 99.5 | 3.4 | 99.3 |
| Fryum | 45.6 | 62.7 | 88.5 | 92.0 | 21.7 | 94.6 |
| Macaroni1 | 20.3 | 22.9 | 70.9 | 98.8 | 36.8 | 98.3 |
| Macaroni2 | 37.7 | 28.8 | 59.3 | 97.8 | 27.5 | 97.6 |
| Pcb1 | 57.8 | 51.6 | 61.2 | 92.7 | 19.8 | 94.1 |
| Pcb2 | 34.7 | 38.4 | 71.6 | 89.7 | 22.9 | 92.4 |
| Pcb3 | 54.6 | 44.6 | 85.3 | 88.4 | 18.0 | 88.4 |
| Pcb4 | 52.1 | 49.9 | 94.4 | 94.6 | 14.0 | 95.7 |
| Pipe_fryum | 58.7 | 44.7 | 75.4 | 96.0 | 29.2 | 98.2 |
| Mean | 46.6 | 47.8 | 79.6 | 94.2 | 24.2 | 95.5 |

Table 20: Fine-grained data-subset-wise performance comparison (PRO) for anomaly segmentation on VisA.

| Object name | CLIP | CLIP-AC | WinCLIP | VAND | CoOp | AnomalyCLIP |
|---|---|---|---|---|---|---|
| Candle | 3.6 | 6.0 | 83.5 | 92.5 | 1.1 | 96.2 |
| Capsules | 15.8 | 22.4 | 35.3 | 86.7 | 18.4 | 78.5 |
| Cashew | 9.6 | 10.9 | 76.4 | 91.7 | 1.7 | 91.6 |
| Chewinggum | 17.8 | 30.2 | 70.4 | 87.3 | 0.1 | 91.2 |
| Fryum | 12.1 | 29.3 | 77.4 | 89.7 | 2.6 | 86.8 |
| Macaroni1 | 8.1 | 13.4 | 34.3 | 93.2 | 18.1 | 89.8 |
| Macaroni2 | 20.9 | 18.4 | 21.4 | 82.3 | 2.7 | 84.2 |
| Pcb1 | 11.7 | 12.5 | 26.3 | 87.5 | 0.1 | 81.7 |
| Pcb2 | 12.8 | 13.9 | 37.2 | 75.6 | 0.7 | 78.9 |
| Pcb3 | 31.7 | 23.6 | 56.1 | 77.8 | 0.0 | 77.1 |
| Pcb4 | 17.1 | 20.3 | 80.4 | 86.8 | 0.0 | 91.3 |
| Pipe_fryum | 16.7 | 6.0 | 82.3 | 90.9 | 0.6 | 96.8 |
| Mean | 14.8 | 17.3 | 56.8 | 86.8 | 3.8 | 87.0 |

Table 21: Fine-grained data-subset-wise performance comparison (AUROC) for anomaly classification on VisA.

| Object name | CLIP | CLIP-AC | WinCLIP | VAND | CoOp | AnomalyCLIP |
|---|---|---|---|---|---|---|
| Candle | 37.9 | 33.0 | 95.4 | 83.8 | 46.2 | 79.3 |
| Capsules | 69.7 | 75.3 | 85.0 | 61.2 | 77.2 | 81.5 |
| Cashew | 69.1 | 72.7 | 92.1 | 87.3 | 75.7 | 76.3 |
| Chewinggum | 77.5 | 76.9 | 96.5 | 96.4 | 84.9 | 97.4 |
| Fryum | 67.2 | 60.9 | 80.3 | 94.3 | 80.0 | 93.0 |
| Macaroni1 | 64.4 | 67.4 | 76.2 | 71.6 | 53.6 | 87.2 |
| Macaroni2 | 65 | 65.7 | 63.7 | 64.6 | 66.5 | 73.4 |
| Pcb1 | 54.9 | 43.9 | 73.6 | 53.4 | 24.7 | 85.4 |
| Pcb2 | 62.6 | 59.5 | 51.2 | 71.8 | 44.6 | 62.2 |
| Pcb3 | 52.2 | 49.0 | 73.4 | 66.8 | 54.4 | 62.7 |
| Pcb4 | 87.7 | 89.0 | 79.6 | 95.0 | 66.0 | 93.9 |
| Pipe_fryum | 88.8 | 86.4 | 69.7 | 89.9 | 80.1 | 92.4 |
| Mean | 66.4 | 65.0 | 78.1 | 78.0 | 62.8 | 82.1 |

Table 22: Fine-grained data-subset-wise performance comparison (AP) for anomaly classification on VisA.

| Object name | CLIP | CLIP-AC | WinCLIP | VAND | CoOp | AnomalyCLIP |
|---|---|---|---|---|---|---|
| Candle | 42.9 | 40.0 | 95.8 | 86.9 | 52.9 | 81.1 |
| Capsules | 81.0 | 84.3 | 90.9 | 74.3 | 85.3 | 88.7 |
| Cashew | 83.4 | 86.1 | 96.4 | 94.1 | 87.1 | 89.4 |
| Chewinggum | 90.4 | 90.2 | 98.6 | 98.4 | 93.1 | 98.9 |
| Fryum | 82.0 | 76.6 | 90.1 | 97.2 | 90.2 | 96.8 |
| Macaroni1 | 56.8 | 58.7 | 75.8 | 70.9 | 52.3 | 86.0 |
| Macaroni2 | 65.0 | 65.8 | 60.3 | 63.2 | 62.2 | 72.1 |
| Pcb1 | 56.9 | 48.4 | 78.4 | 57.2 | 36.0 | 87.0 |
| Pcb2 | 63.2 | 59.8 | 49.2 | 73.8 | 47.3 | 64.3 |
| Pcb3 | 53.0 | 47.6 | 76.5 | 70.7 | 54.8 | 70.0 |
| Pcb4 | 88.0 | 90.6 | 77.7 | 95.1 | 66.3 | 94.4 |
| Pipe_fryum | 94.6 | 93.7 | 82.3 | 94.8 | 89.7 | 96.3 |
| Mean | 71.5 | 70.1 | 81.2 | 81.4 | 68.1 | 85.4 |

Table 23: Fine-grained data-subset-wise performance comparison (AUROC) for anomaly segmentation on MPDD.

| Object name | CLIP | CLIP-AC | WinCLIP | VAND | CoOp | AnomalyCLIP |
|---|---|---|---|---|---|---|
| Bracket_black | 85.3 | 86.4 | 57.8 | 96.3 | 9.3 | 95.7 |
| Bracket_brown | 26.9 | 31.5 | 72.2 | 86.2 | 20.2 | 94.4 |
| Bracket_white | 83.5 | 77.4 | 79.5 | 99.0 | 8.3 | 99.8 |
| Connector | 56.5 | 52.9 | 79.0 | 90.6 | 7.6 | 97.2 |
| Metal_plate | 64.3 | 52.5 | 92.6 | 93.1 | 14.1 | 93.8 |
| Tubes | 56.4 | 51.5 | 77.6 | 99.1 | 33.2 | 98.1 |
| Mean | 62.1 | 58.7 | 76.4 | 94.1 | 15.4 | 96.5 |

Table 24: Fine-grained data-subset-wise performance comparison (PRO) for anomaly segmentation on MPDD.

| Object name | CLIP | CLIP-AC | WinCLIP | VAND | CoOp | AnomalyCLIP |
|---|---|---|---|---|---|---|
| Bracket_black | 62.6 | 58.9 | 43 | 89.7 | 1.5 | 85.2 |
| Bracket_brown | 2.8 | 4.0 | 25.0 | 70.3 | 0.4 | 77.7 |
| Bracket_white | 47.9 | 41.6 | 57.6 | 93.1 | 0.0 | 98.8 |
| Connector | 22.8 | 20.2 | 44.6 | 74.5 | 0.0 | 89.8 |
| Metal_plate | 31.5 | 27.0 | 78.2 | 74.5 | 0.2 | 86.9 |
| Tubes | 30.4 | 22.9 | 44.7 | 96.9 | 11.5 | 93.6 |
| Mean | 33.0 | 29.1 | 48.9 | 83.2 | 2.3 | 88.7 |

Table 25: Fine-grained data-subset-wise performance comparison (AUROC) for anomaly classification on MPDD.

| Object name | CLIP | CLIP-AC | WinCLIP | VAND | CoOp | AnomalyCLIP |
|---|---|---|---|---|---|---|
| Bracket_black | 32.4 | 32.8 | 41.5 | 66.1 | 36.9 | 67.3 |
| Bracket_brown | 50.9 | 57.9 | 48.6 | 64.0 | 43.9 | 62.2 |
| Bracket_white | 45.4 | 42.6 | 40.2 | 79.6 | 48.9 | 64.9 |
| Connector | 75 | 76.2 | 79.3 | 78.8 | 38.3 | 86.9 |
| Metal_plate | 34.9 | 54.8 | 93.4 | 53.8 | 77.0 | 85.2 |
| Tubes | 87.3 | 72.8 | 78.7 | 95.9 | 85.4 | 95.5 |
| Mean | 54.3 | 56.2 | 63.6 | 73.0 | 55.1 | 77.0 |

Table 26: Fine-grained data-subset-wise performance comparison (AP) for anomaly classification on MPDD.

| Object name | CLIP | CLIP-AC | WinCLIP | VAND | CoOp | AnomalyCLIP |
|---|---|---|---|---|---|---|
| Bracket_black | 47.8 | 48.6 | 56.9 | 71.7 | 50.0 | 72.9 |
| Bracket_brown | 66.2 | 72.0 | 69.5 | 79.0 | 65.7 | 80.8 |
| Bracket_white | 51.2 | 47.3 | 45.1 | 82.3 | 57.5 | 68.5 |
| Connector | 62.2 | 61.4 | 61.3 | 71.8 | 26.4 | 76.8 |
| Metal_plate | 70.6 | 78.5 | 97.6 | 78.3 | 92.0 | 94.7 |
| Tubes | 94.4 | 88.2 | 89.1 | 98.1 | 93.6 | 98.1 |
| Mean | 65.4 | 66.0 | 69.9 | 80.2 | 64.2 | 82.0 |

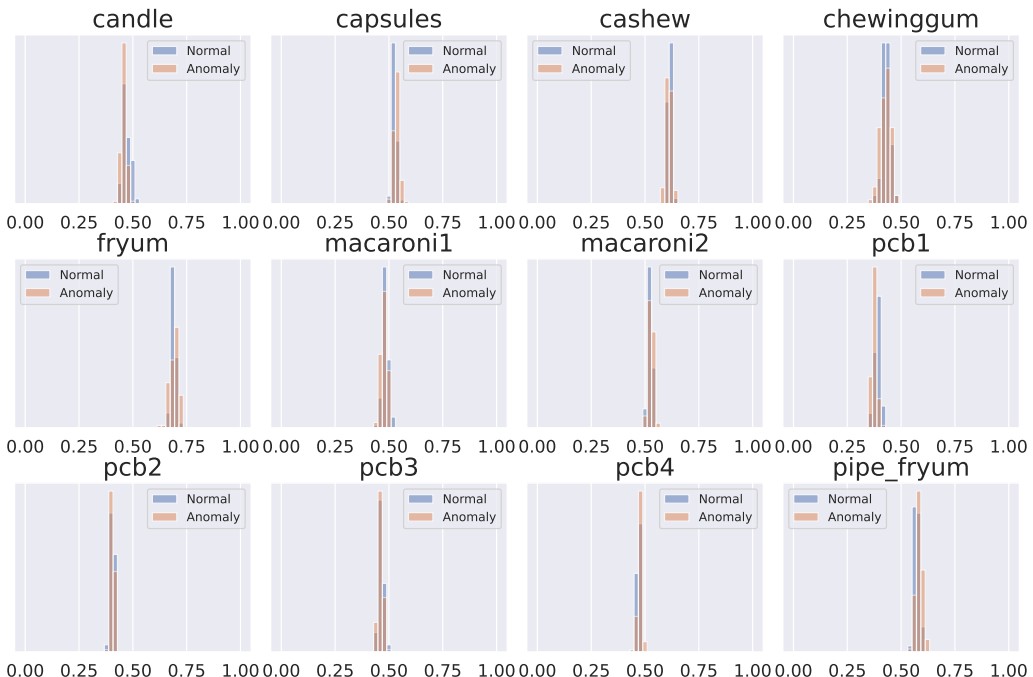

Figure 11: Similarity scores of CLIP on VisA. Each sub-figure represents the visualization of one object.

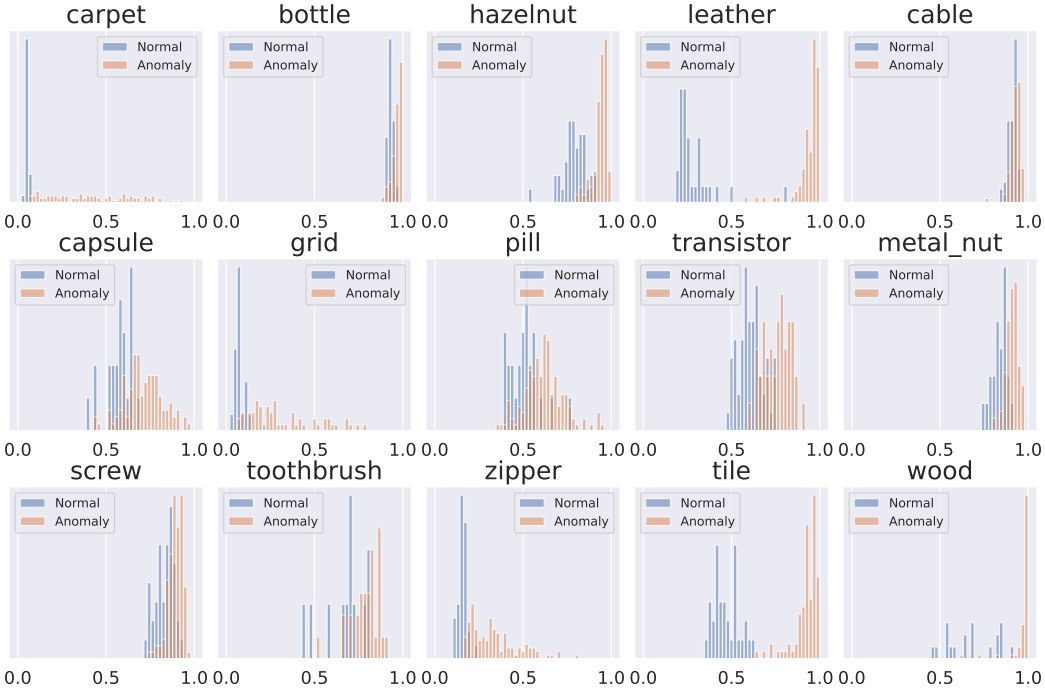

Figure 12: Similarity scores of AnomalyCLIP on MVTec AD. Each sub-figure represents the visualization of one object.

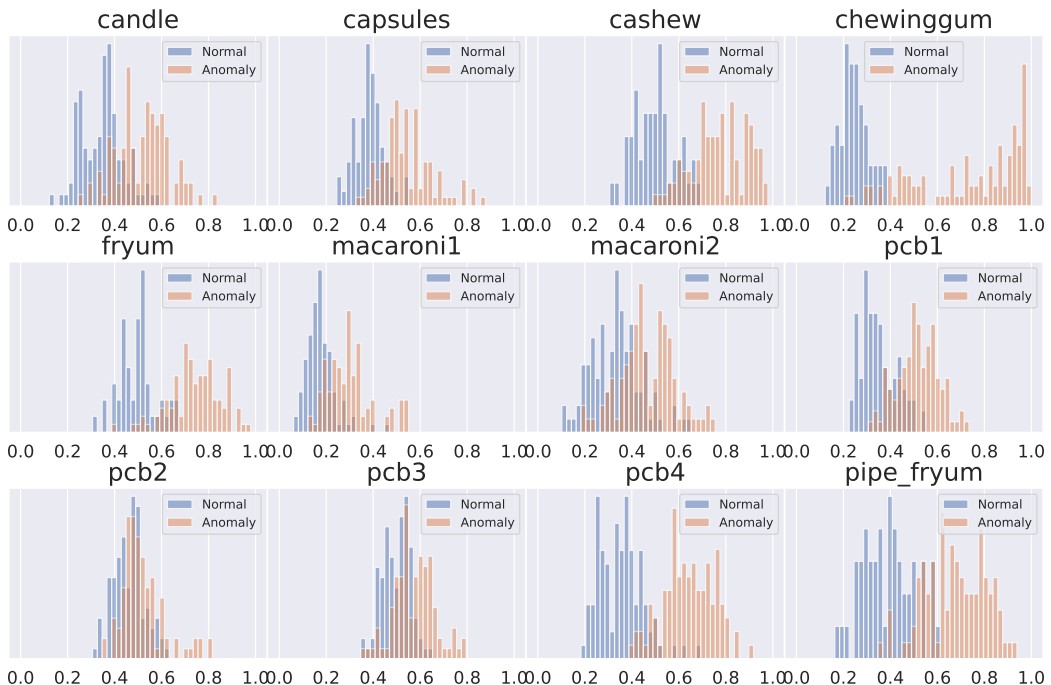

Figure 13: Similarity scores of AnomalyCLIP on VisA. Each sub-figure represents the visualization of one object.

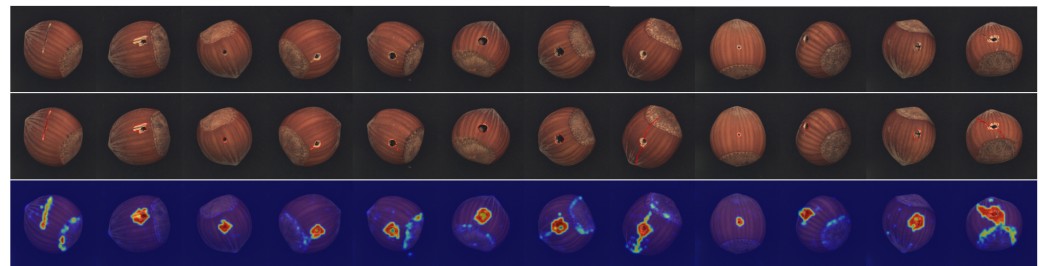

Figure 14: Anomaly score maps for the data subset, hazelnut, in MVTec AD. The first row represents the input, and we circle the anomaly regions in the second row. The last row presents the segmentation results from AnomalyCLIP.

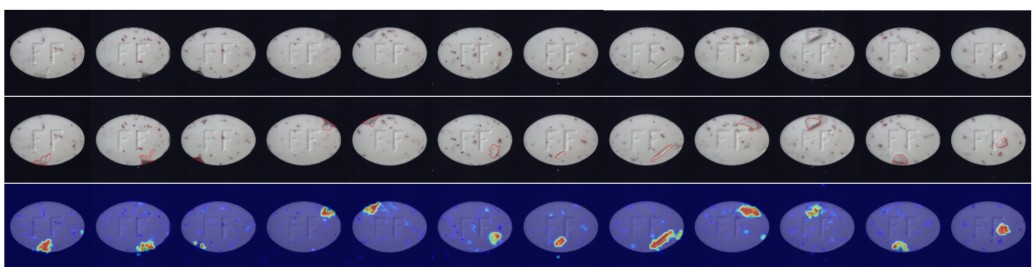

Figure 15: Anomaly score maps for the data subset, pill, in MVTec AD. The first row represents the input, and we circle the anomaly regions in the second row. The last row presents the segmentation results from AnomalyCLIP.

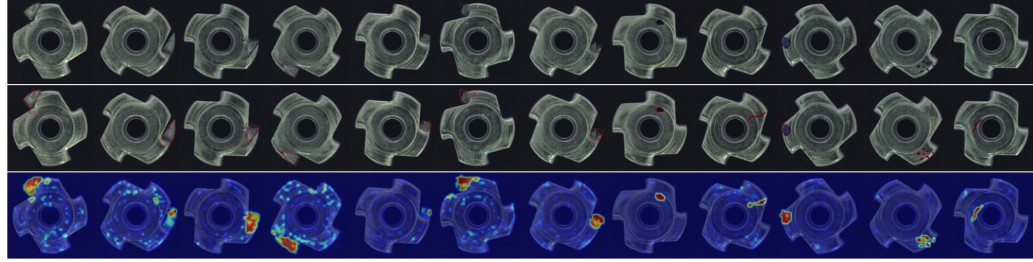

Figure 16: Anomaly score maps for the data subset, metal nut, in MVTec AD. The first row represents the input, and we circle the anomaly regions in the second row. The last row presents the segmentation results from AnomalyCLIP.

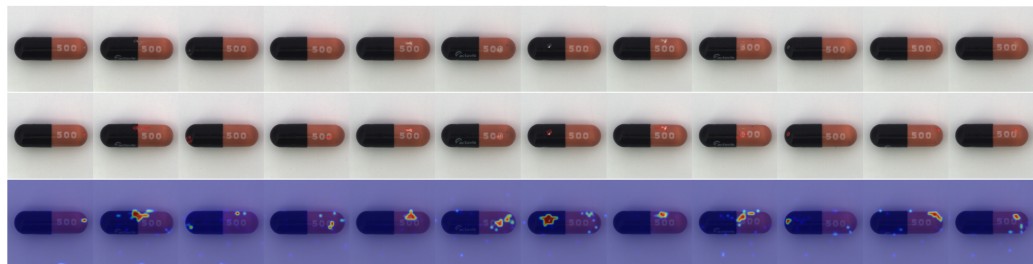

Figure 17: Anomaly score maps for the data subset, capsule, in MVTec AD. The first row represents the input, and we circle the anomaly regions in the second row. The last row presents the segmentation results from AnomalyCLIP.

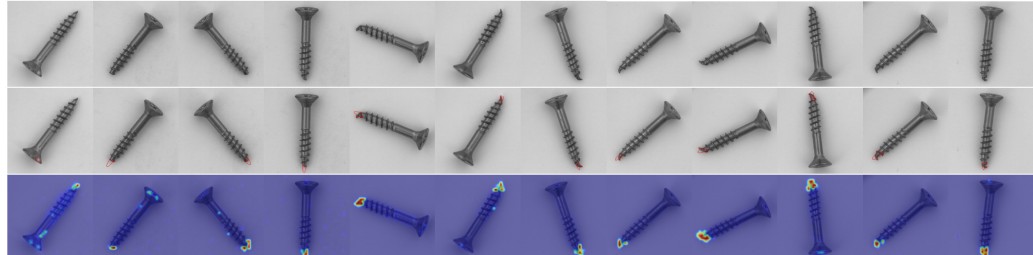

Figure 18: Anomaly score maps for the data subset, screw, in MVTec AD. The first row represents the input, and we circle the anomaly regions in the second row. The last row presents the segmentation results from AnomalyCLIP.

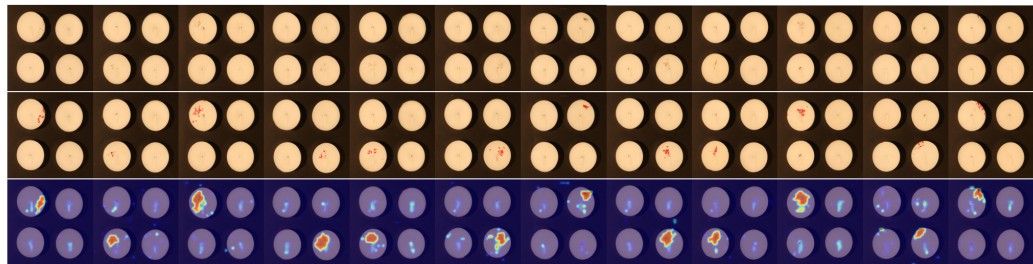

Figure 19: Anomaly score maps for the data subset candle. The first row represents the input, and we circle the anomaly regions in the second row. The last row presents the segmentation results from AnomalyCLIP.

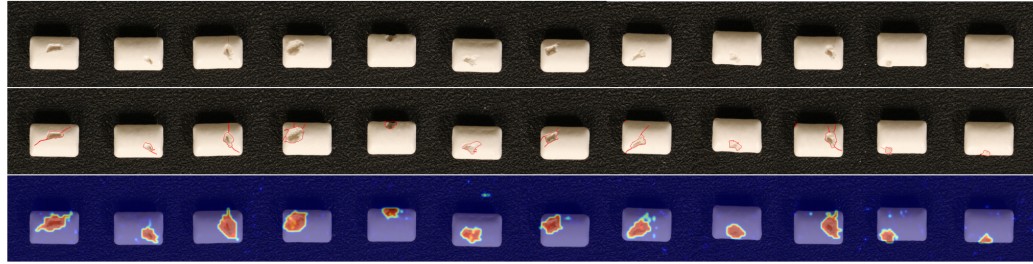

Figure 20: Anomaly score maps for the data subset chewinggum. The first row represents the input, and we circle the anomaly regions in the second row. The last row presents the segmentation results from AnomalyCLIP.

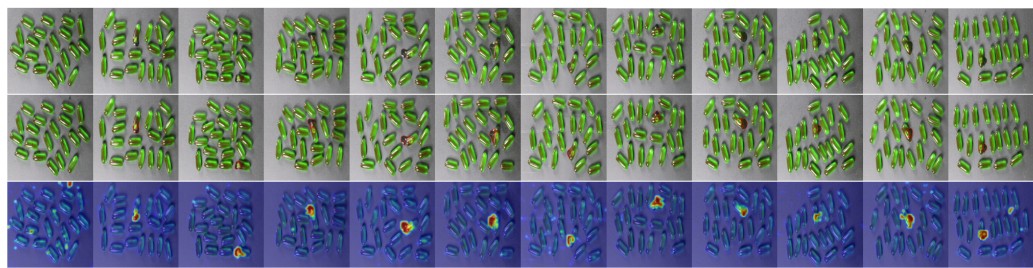

Figure 21: Anomaly score maps for the data subset capusle. The first row represents the input, and we circle the anomaly regions in the second row. The last row presents the segmentation results from AnomalyCLIP.

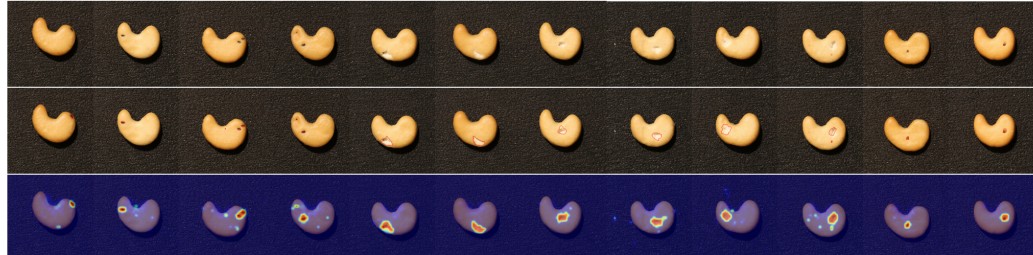

Figure 22: Anomaly score maps for the data subset cashew. The first row represents the input, and we circle the anomaly regions in the second row. The last row presents the segmentation results from AnomalyCLIP.

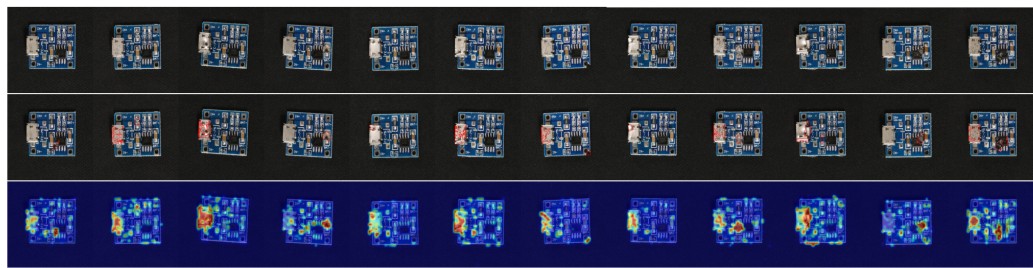

Figure 23: Anomaly score maps for the data subset pcb. The first row represents the input, and we circle the anomaly regions in the second row. The last row presents the segmentation results from AnomalyCLIP.

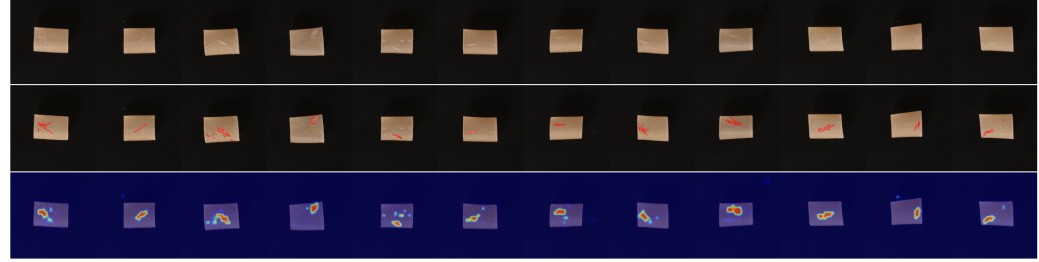

Figure 24: Anomaly score maps for the data subset pip fryum. The first row represents the input, and we circle the anomaly regions in the second row. The last row presents the segmentation results from AnomalyCLIP.

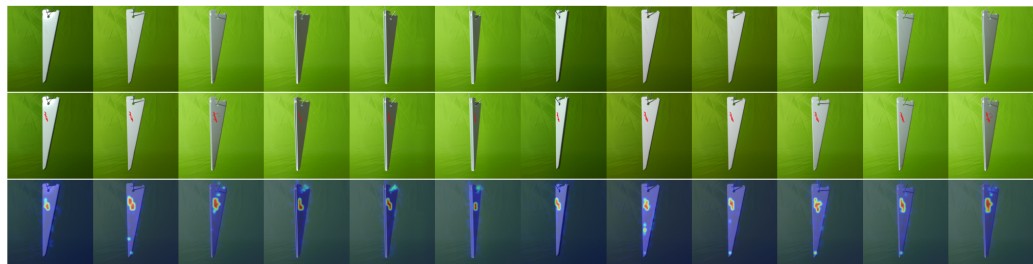

Figure 25: Similarity scores for the data subset bracket. The first row represents the input, and we circle the anomaly regions in the second row. The last row presents the segmentation results from AnomalyCLIP.

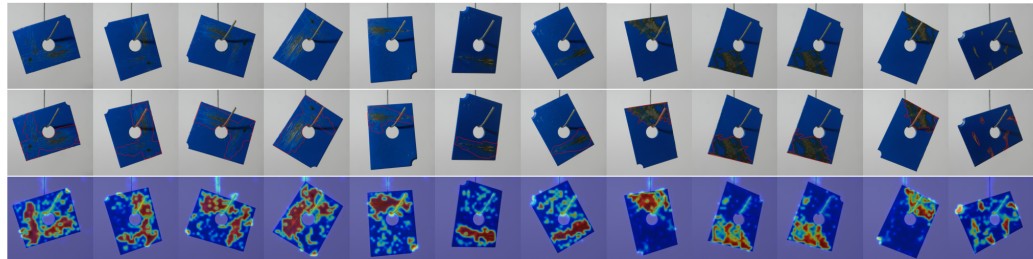

Figure 26: Anomaly score maps for the data subset metal plate. The first row represents the input, and we circle the anomaly regions in the second row. The last row presents the segmentation results from AnomalyCLIP.

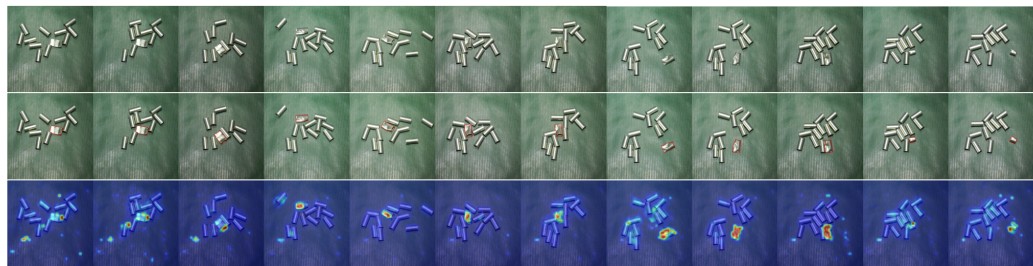

Figure 27: Anomaly score maps for the data subset tube. The first row represents the input, and we circle the anomaly regions in the second row. The last row presents the segmentation results from AnomalyCLIP.

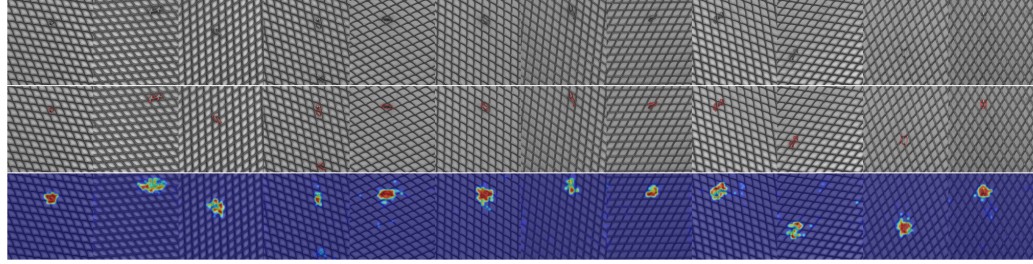

Figure 28: Anomaly score maps for the data subset grid. The first row represents the input, and we circle the anomaly regions in the second row. The last row presents the segmentation results from AnomalyCLIP.

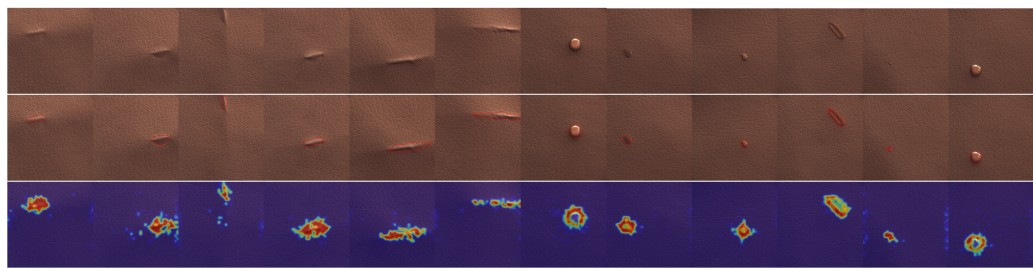

Figure 29: Anomaly score maps for the data subset leather. The first row represents the input, and we circle the anomaly regions in the second row. The last row presents the segmentation results from AnomalyCLIP.

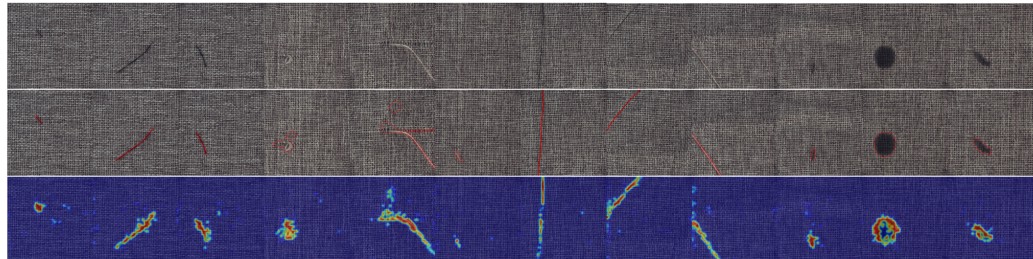

Figure 30: Anomaly score maps for the data subset carpet. The first row represents the input, and we circle the anomaly regions in the second row. The last row presents the segmentation results from AnomalyCLIP.

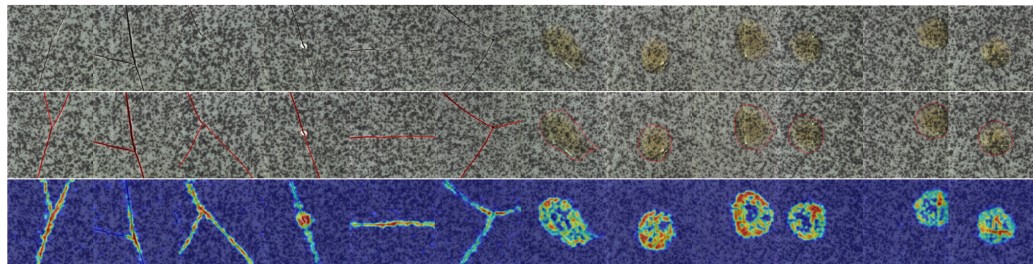

Figure 31: Anomaly score maps for the data subset tile. The first row represents the input, and we circle the anomaly regions in the second row. The last row presents the segmentation results from AnomalyCLIP.

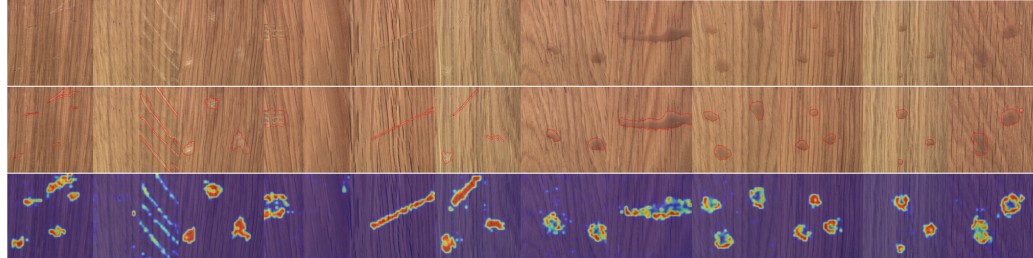

Figure 32: Anomaly score maps for the data subset wood. The first row represents the input, and we circle the anomaly regions in the second row. The last row presents the segmentation results from AnomalyCLIP.

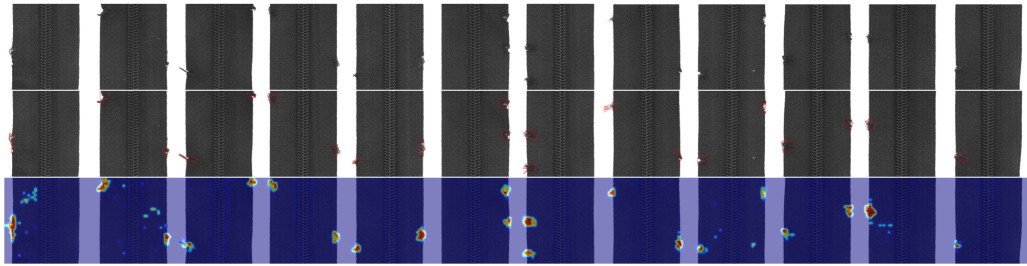

Figure 33: Anomaly score maps for the data subset zipper. The first row represents the input, and we circle the anomaly regions in the second row. The last row presents the segmentation results from AnomalyCLIP.

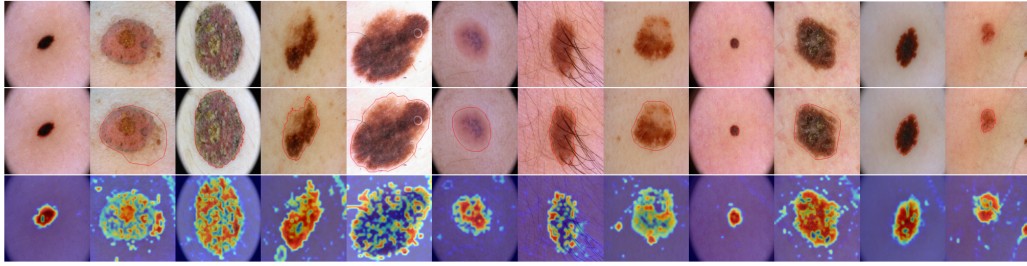

Figure 34: Similarity scores for the data subset skin.

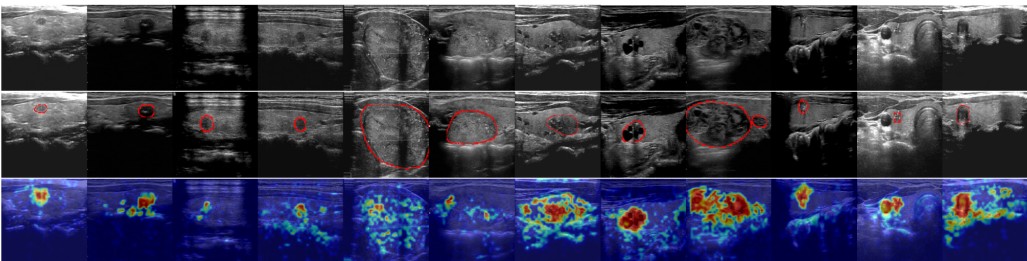

Figure 35: Anomaly score maps for the data subset thyroid. The first row represents the input, and we circle the anomaly regions in the second row. The last row presents the segmentation results from AnomalyCLIP.

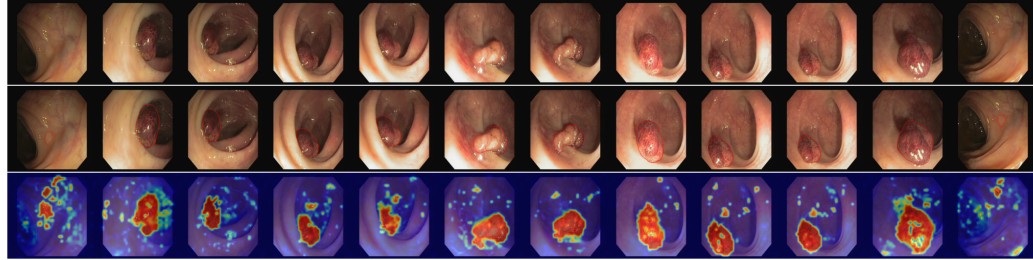

Figure 36: Anomaly score maps for the data subset colon. The first row represents the input, and we circle the anomaly regions in the second row. The last row presents the segmentation results from AnomalyCLIP.

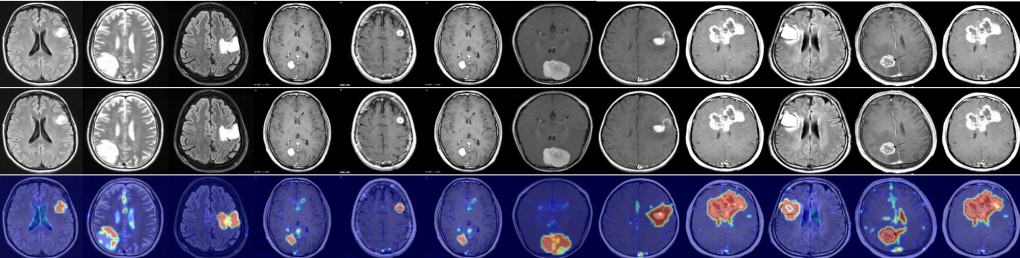

Figure 37: Anomaly score maps for the data subset brain. The first row represents the input, and we circle the anomaly regions in the second row. The last row presents the segmentation results from AnomalyCLIP.

