# OpenReview forum: "AnomalyCLIP: Object-agnostic Prompt Learning for Zero-shot Anomaly Detection"
_ICLR.cc/2024/Conference — ICLR 2024 poster_

### Official Review · Reviewer_5wSC · 2023-10-15

**Soundness:** 3 good
**Presentation:** 4 excellent
**Contribution:** 4 excellent
**Rating:** 8
**Confidence:** 3

**Summary:**

This paper proposes a zero-shot anomaly detection framework using CLIP and object-agnostic text prompts with learnable prompt tokens and a modified attention mechanism.

**Strengths:**

The paper is well written and easy to follow overall

Experiments are comprehensive, with many datasets and evaluation metrics, as well as the ablation study

VLMs for anomaly detection is relatively under-explored and prompt learning is especially new amongst AD literature.

**Weaknesses:**

Little/no discussion about the additional cost of training

No comparison of different ViT backbones of CLIP

Figure 6 feels confusing and unnecessary

It would be better to provide more intuition why the Focal and Dice loss are necessary

Figure 2 caption should be more explanatory

**Questions:**

Besides all of the other modifications such as DPAM, why are object-agnostic prompts better than object-aware prompts for a specific object class?

Is it true that hidden layer visual embeddings are only used for the segmentation map, while the final output embedding is only used for the image-level score?

---

> ### Author Response · Authors · 2023-11-22
> **Response to Reviewer 5wSC (Part 1)**
>
> We thank the reviewer for the positive review and encouraging comments.
>
> **Q1: Little/no discussion about the additional cost of training**
>
> We have supplemented the analysis about the additional cost of training by measuring the time taken during training (training time per epoch) in Table 8 of Appendix D. For a fair comparison, all experiments are conducted in a single NVIDIA RTX 3090 24GB GPU, and the GPU is kept free before evaluation. In Table 8, AnomalyCLIP takes 13.71 min per epoch on MVTec AD (The total number of samples is 1725) and only requires a total of 15 epochs for the whole fine-tuning. Once AnomalyCLIP finishes fine-tuning, AnomalyCLIP can be applied to different datasets and domains without additional training. We also compare AnomalyCLIP with other baselines that need auxiliary data (i.e., CoOp and VAND). For example, the minimum training time per epoch is 12.25 min of CoOp, and hence, AnomalyCLIP has a similar training computation cost as CoOp.
>
> **Q2: No comparison of different ViT backbones of CLIP**
>
> We have added the backbone ablation study in Table 13 in Appendix D. We originally used ViT-L-14 as the backbones of CLIP, which is a representative backbone in CLIP. To fully explore the effect of different sizes of backbones, we use ViT-B-16-plus and ViT-H-14, which is a smaller and larger model than ViT-L-14, respectively. Compared to pixel-level performance 90.5% AUROC and 90.8% AUROC from ViT-B-16-plus and ViT-H-14, ViT-L-14 achieves the best performance 91.1% AUROC. The results indicate that AnomalyCLIP maintains promising performance across various ViT backbones. Meanwhile, it also reveals that increasing the model size does not necessarily result in improved performance. The larger model may introduce a strong bias towards object semantics, hindering AnomalyCLIP's ability to learn object-agnostic text prompts capturing generic normality and abnormality in an image. However, a smaller backbone like ViT-B-16-plus could cause AnomalyCLIP to struggle with generalization to unseen anomalies.
>
> **Q3: Figure 6 feels confusing and unnecessary**
>
> Since CLIP is pre-trained to align the textual semantics and the corresponding object, it harms the representation of local visual semantics which is important to anomaly localization. Therefore, AnomalyCLIP proposes DPAM to improve the local visual semantics. The DPAM strategy has three variants, i.e., $Q$-$Q$, $K$-$K$, and $V$-$V$ attention. To investigate the effect of different variants, we present their respective ZSAD performance in Figure 6. We agree that these results are not as important as the other results in justifying the effectiveness of AnomalyCLIP; they are presented to provide an insight into the effectiveness of the attention mechanism in CLIP on ZSAD since CLIP has emerged to be one key tool to facilitate this task.
>
> **Q4: It would be better to provide more intuition why the Focal and Dice loss are necessary**
>
> We thank the reviewer for pointing this out. Although local features are refined by DPAM, there still exists misalignment among textual semantic and local visual semantics, especially for unseen objects. Focal and Dice loss play a crucial role in optimizing local context. They are introduced to empower object-agnostic text prompts to focus on fine-grained, local abnormal regions from intermediate layers of the visual encoder. As mentioned in Section 3.2, Focal loss addresses the imbalance between anomaly and normal pixels, typically caused by the smaller size of anomalous regions. Meanwhile, Dice loss aims to precisely constrain the anomaly boundary by measuring the overlap between the predicted segmentation ($S_n$/$S_a$) and the ground truth mask. To provide a more comprehensive analysis, we have included an ablation study on Focal and Dice loss in Table 14 of Appendix D. Compared to scenarios without local context optimization, Dice loss improves the pixel-level and image-level performance from 80.3%AUROC to 87.2%AUROC and 86.6%AUROC to 90.1%AUROC in pixel level on MVTec AD and VisA. Focal loss also brings the performance gain of 10.3%AUROC and 8.3%AUROC. Combining Focal and Dice loss, AnomalyCLIP achieves the best results (i.e., 91.1%AUROC and 95.5%AUROC). Note that the global context optimization is always used during the ablation, since we need at least one loss function to drive the optimization.
>
> **Q5: Figure 2 caption should be more explanatory**
>
> Thank you for your effort on helping improve the paper. We have added more illustrations in the revised version: "To adapt CLIP to ZSAD, AnomalyCLIP introduces object-agnostic text prompt templates to capture generic normality and abnormality regardless of the object semantics. Then, we introduce glocal context optimization to incorporate global and fine-grained anomaly semantics into object-agnostic textual embedding learning. Finally, textual prompt tuning and DPAM are used to support the learning in the textual and local visual spaces of CLIP.\"

---

> > ### Author Response · Authors · 2023-11-22
> > **Response to Reviewer 5wSC (Part 2)**
> >
> > **Q6: Besides all of the other modifications such as DPAM, why are object-agnostic prompts better than object-aware prompts for a specific object class?**
> >
> > Thank you for raising this question. The object semantics in the object-aware text prompt can be regarded as the condition information. During fine-tuning, anomalies across different objects are used to ensure that the prompt captures generic anomaly patterns. In such a process, the object-aware prompt actually learns the condition information of specific objects, hindering the prompt from learning the generic abnormality for unseen objects. Different from the object-aware text prompt, the object-agnostic text prompt excludes the object semantics from text prompt templates, allowing learnable text prompt templates to focus on capturing the characteristics of anomalies themselves, rather than the objects. Therefore, when detecting anomalies in unseen objects, the object-agnostic text prompt can better capture the underlying anomaly patterns than the object-aware text prompt.
> >
> > **Q7: Is it true that hidden layer visual embeddings are only used for the segmentation map, while the final output embedding is only used for the image-level score?**
> >
> > Thank you for your detailed feedback. Besides the intermediate local visual embeddings used for segmentation, we also include the local visual embeddings of the final output. As for image classification, we only use the global visual embedding of the final output for image-level score. For clear clarification, the $k$-th visual representation of the visual encoder is denoted as $F_k\in R ^ {(1 + H\times W) \times D}$, where the class token $f_k \in R ^ D$ is treated as its global visual embedding, and patch tokens $f_k^m \in R ^ {H\times W\times D}$ are referred to as local visual embeddings. In AnomalyCLIP, $k$ is set to {6, 12, 18, 24}, and the 24-th layer is the top layer of the visual encoder. To combine more local visual semantics, we integrate $f_{6}^m$, $f_{12}^m$, $f_{18}^m$, and $f_{24}^m$ to perform anomaly segmentation. Since anomaly classification requires accurate global representation, we only use the class token of the top layer, $f_{24}$, to perform anomaly classification. We hope this helps clarify the confusion.

---

### Official Review · Reviewer_vcXy · 2023-10-19

**Soundness:** 3 good
**Presentation:** 3 good
**Contribution:** 3 good
**Rating:** 5
**Confidence:** 5

**Summary:**

This paper targets ZSAD leveraging visual language pretrained models. To learn object-agnostic information, this paper proposes to use the well-known text prompt technique, allowing the model to focus on the abnormal image regions rather than the object semantics. Experiments are conducted on several anomaly detection benchmarks.

**Strengths:**

1.	The contribution is clear, with well-organized paper architecture.
2.	The motivation sounds reasonable.
3.	Experiments show the effectiveness of the proposed method.

**Weaknesses:**

1. The motivation behind DPAM appears to be relatively weak. It is not entirely clear why employing V-V self-attention would preserve more localized information compared to the original Q-K attention. Relying solely on a single instance shown in Figure 3 may not furnish sufficient evidence to support this assertion. Furthermore, the rationale for treating these features separately remains unclear. What might be the outcome if we were to combine Q-Q, K-K, V-V, and Q-K features into an ensemble?

2. The ablation studies lack clarity regarding the specific details of each component, making it challenging to assess the actual contributions of the proposed method:
(i) The experiments in Table 4 involve the incremental addition of modules (T1-T4), but a more informative approach would be to examine the performance when each of them is removed. For instance, it is unclear how DPAM (T1) performs when used in conjunction with both T2, T3, and T4. So an importment setting should be removing T1 but keeping using T2-T4. Additionally, the increase in AUC from 47.9 to 54.8 by introducing T1 does not signify a significant improvement (both values are close to random guessing) for a two-class classification problem.
(ii) In Table 5, it remains unclear how the models are optimized when both the global and local losses are absent.

**Questions:**

See the weakness.

---

> ### Author Response · Authors · 2023-11-22
> **Response to Reviewer vcXy (Part 1)**
>
> Thank you for your review and insightful comments.
>
> **Q1: The motivation behind DPAM appears to be relatively weak. It is not entirely clear why employing V-V self-attention would preserve more localized information compared to the original Q-K attention.**
>
> Thank you for raising this concern. Please refer to our **General Response** above for an intuitive explanation of DPAM strategy from the attention mechanism.
>
> **Q2: Relying solely on a single instance shown in Figure 3 may not furnish sufficient evidence to support this assertion.**
>
> We have added experiments to study the effect of DPAM in Table 7 of Appendix D. Without our proposed DPAM module, AnomalyCLIP shows a decrease from 91.1% AUROC to 87.9% AUROC in pixel-level performance and from 91.5% AUROC to 80.7% AUROC in image-level performance on MVTec AD. Additionally, there is a decrease from 95.5% AUROC to 91.9% AUROC in pixel-level performance and from 82.1% AUROC to 73.0% AUROC in image-level performance on VisA. The results demonstrate the superiority of $V$-$V$ attention over $Q$-$K$ attention. However, the performance decreases at the image level is more pronounced than that at the pixel level. This discrepancy is attributed to the fact that the total loss places greater emphasis on local context optimization, driven by a larger local loss compared to the case with DPAM. As a result, AnomalyCLIP lacks sufficient optimization for global alignment, leading to a significant decline in image-level performance. Note that the original CLIP (no DPAM) exploits $Q$-$K$ features and suffers from weak segmentation performance.
>
> **Q3: What might be the outcome if we were to combine Q-Q, K-K, V-V, and Q-K features into an ensemble?**
>
> Thank you for your helpful comments. Ensembling different features is an excellent idea. We have added two experiments to explore two ensemble methods, namely AnomalyCLIP$\_{ensemble1}$ and AnomalyCLIP$\_{ensemble2}$, involving the ensemble of **Q-Q, K-K, and V-V** and the ensemble of **Q-K, Q-Q, K-K, and V-V**, respectively in Table 12 in Appendix D. We average the logit output of different features for the ensemble. In Table 12, AnomalyCLIP$\_{ensemble1}$ shows performance improvement by leveraging the advantages of three DPAM features. However, while AnomalyCLIP$\_{ensemble2}$ outperforms the $Q$-$K$ feature version, it experiences a performance decrease compared to $V$-$V$ from 91.1%AUROC to 90.7%AUROC in pixel level and 91.5%AUROC to 90.8%AUROC in image level on MVTec AD. There is also a decline from 95.5%AUROC to 94.9%AUROC in pixel level and 82.1%AUROC to 80.7%AUROC in image level on VisA. The decline in performance upon adding $Q$-$K$ features to AnomalyCLIP$_{ensemble1}$ suggests that the $Q$-$K$ feature fails to provide valid local visual semantics to facilitate ZSAD. Note that the original CLIP exploits $Q$-$K$ features and gets weak segmentation performance. The good pixel-level performance of $Q$-$K$ in AnomalyCLIP is attributed to local optimization, where the object-agnostic prompt helps alleviate the disrupted local visual semantics of $Q$-$K$.

---

> > ### Author Response · Authors · 2023-11-22
> > **Response to Reviewer vcXy (Part 2)**
> >
> > **Q4: A more informative approach would be to examine the performance when each of them is removed.**
> >
> > Thank you for detailed feedback. To demonstrate the effectiveness of each module we propose, we have supplemented the additional module ablation in Table 7 of Appendix D. We test the contribution of one module by removing one module and maintaining the rest module.
> >
> > 1. The effectiveness of DPAM ($T_1$). When we remove DPAM, the results show a decrease from 91.1% AUROC to 87.9% AUROC in pixel-level performance and from 91.5% AUROC to 80.7% AUROC in image-level performance. This performance decline indicates the importance of DPAM, which enhances local visual semantics by modifying the attention mechanism. However, the decrease in performance at the image level is more pronounced than that at the pixel level. This discrepancy is attributed to the fact that the total loss places greater emphasis on local context optimization, driven by a larger local loss compared to the case with DPAM.
> > 2. The effectiveness of object-agnostic prompt learning ($T_2$). Excluding object-agnostic prompt learning makes AnomalyCLIP suffer from the huge performance gap (i.e., 91.1% AUROC to 84.3% AUROC in pixel-level and 91.5% AUROC to 65.6% AUROC in image-level). This performance decline illustrates that the object-agnostic text prompt template plays a significant role in improving the performance of AnomalyCLIP at both pixel and image levels.
> > 3. The effectiveness of textual prompt tuning ($T_3$). When removing textual prompt tuning, the performance of AnomalyCLIP declines from 91.1% AUROC to 90.6% AUROC in pixel-level performance and from 91.5% AUROC to 90.4% AUROC in image-level performance. This demonstrates the importance of adapting original textual space by adding learnable textual tokens in the text encoder.
> > 4. The effectiveness of the integration of multi-layer local visual semantics ($T_4$). When removing multi-layer local visual semantics, the outcomes reveal a decrease from 91.1% AUROC to 90.0% AUROC in pixel-level performance and from 91.5% AUROC to 91.0% AUROC in image-level performance. This performance decline indicates the importance of incorporating multi-layer local visual semantics.
> >
> > **Q5: In Table 5, it remains unclear how the models are optimized when both the global and local losses are absent.**
> >
> > Thank you for pointing this out. When both the global and local losses are absent, the model is not optimized. Therefore, we report the results of the base method, where there is no $T_1$, $T_2$, $T_3$, and $T_4$. However, it would be better to contain $T_1$ and $T_4$ for appropriate analysis. This is because $T_1$ and $T_4$ do not require optimization, while $T_2$ and $T_3$ involve the optimization process. We have updated the results in the revised paper.

---

### Official Review · Reviewer_GzuZ · 2023-10-27

**Soundness:** 2 fair
**Presentation:** 3 good
**Contribution:** 3 good
**Rating:** 6
**Confidence:** 5

**Summary:**

The paper proposes a zero-shot anomaly detection/segmentation method utilizing CLIP and prompt learning. The authors observe that CLIP-like VLMs learn class semantic features of the object and pay less attention to normality/abnormality, e.g., associated with fine-grained local features. They assume that the normality/abnormality features can be object-agnostic. For adapting the pretrained CLIP to focus on normality and abnormality features, they proposed to learn two text prompts for each respectively. To learn these generic prompts, they utilize both global optimization, i.e., maximizing similarity between image embeddings and text embeddings, and local optimization, i.e., maximizing similarity between image-patch embeddings and text embeddings. They also show that adding more learnable tokens to the text encoder and image encoder of CLIP can improve results, and using V-V attention can help in refining the local features. The paper provides a comprehensive empirical evaluation of 17 real-world datasets showcasing the effectiveness and advantages of the proposed method over existing zero-shot anomaly detection baselines.

**Strengths:**

1. The paper brings the principle of prompt learning for CLIP-like VLMs in zero-shot anomaly detection.
2. The proposed method achieves remarkable zero-shot anomaly detection performance on both industrial anomaly detection datasets and medical datasets.
3. The paper provides a large body of ablation studies for each involved design component individually.

**Weaknesses:**

1.  My largest worry is the paper relies on an optimistic assumption of abnormality, namely, the abnormality is object-agnostic.  The assumption might hold in some cases. The learned model is able to detect the abnormality that has similar abnormal patterns to the features in the auxiliary training set, e.g., detecting the defects of a manufactured part. However, the assumption can also fail in some cases. For example, the model may misclassify a product with texture patterns similar to scratches as an abnormal one, even though the scratch-like textures are normal texture patterns of the product.  It would be better to explicitly discuss the successful cases and the potential failures of the proposed method.

2. Minor comment: $S_n$ and $S_a$ are first-time used in $L_{local}$ on page 5, but are clearly defined on page 6. Having the definitions and equations of $S_n$ and $S_a$ before $L_{local}$ would be good.

3. The discussion of some related work about zero-shot anomaly detection, e.g., [1,2,3], is missing.

[1] Philipp Liznerski, Lukas Ruff, Robert A Vandermeulen, Billy Joe Franks, Klaus Robert Muller, and Marius Kloft. Exposing outlier exposure: What can be learned from few, one, and zero outlier images. Transactions on Machine Learning Research, 2022.

[2] Sepideh Esmaeilpour, Bing Liu, Eric Robertson, and Lei Shu. Zero-shot out-of-distribution detection based on the pretrained model clip. In Proceedings of the AAAI conference on artificial intelligence, 2022

[3] Aodong Li, Chen Qiu, Marius Kloft, Padhraic Smyth, Maja Rudolph, Stephan Mandt. Zero-Shot Batch-Level Anomaly Detection. Thirty-seventh Conference on Neural Information Processing Systems, 2023.

**Questions:**

1. Why is learning two text prompts necessary? Do you have comparisons to $g_n = [V_1][V_2]...[V_E][object]$ and $g_a = [V_1][V_2]...[V_E][damaged][object]$?
1. How is the performance of the variant with all components except DPAM?

---

> ### Author Response · Authors · 2023-11-22
> **Response to Reviewer GzuZ (Part 1)**
>
> Thank you for your positive review and the insightful comments.
>
> **Q1: The model may misclassify a product with texture patterns similar to scratches as an abnormal one, even though the scratch-like textures are normal texture patterns of the product. It would be better to explicitly discuss the successful cases and the potential failures of the proposed method.**
>
> Thank you for raising this concern. We agree that in a minority of cases, AnomalyCLIP exhibits false detection due to the object-agnostic assumption. However, ZSAD requires detection models to detect anomalies without any training sample in a target dataset, and capturing the object-agnostic anomaly patterns is a promising solution because anomalies from different application scenarios typically have substantial variations in their visual appearance, foreground objects, and background features. As suggested by the reviewer, we have supplemented the experiment to explicitly discuss the successful cases and the potential failures in Figure 7 in Appendix D. In Figure 7(a), AnomalyCLIP accurately detects scratch-like patterns on the product, even when they typically appear in the texture. However, false detection occurs when scratch-like patterns are situated in the background, as depicted in Figure 7(b). Meanwhile, we also show the color stain patterns. As shown in Figure 7(c), AnomalyCLIP successfully detects the color stain, which exhibits subtle visual differences from the detected entities. On the other hand, AnomalyCLIP may face challenges when the normal region displays patterns that are almost indistinguishable to the naked eye from anomalies. For instance, in skin cancer detection, the normal regions are falsely detected as anomalies are visually similar to the disease region in Figure 7(d). Also, the stain interference in the background is a problem. These failure cases illustrate that the importance of mitigating background interference and achieving fine-grained discrimination, especially in cases of visually similar abnormalities. Exploring these challenges for enhancing ZSAD is a valuable direction for future research.
>
> **Q2: Minor comment: $S_n$ and $S_a$ are first-time used in $L_{local}$ on page 5, but are clearly defined on page 6. Having the definitions and equations of $S_n$ and $S_a$ before $L_{local}$ would be good.**
>
> Thank you for your effort on helping improve our paper. We have updated the definitions of $S_n$ and $S_a$ before $L_{local}$.
>
> **Q3: The discussion of some related work about zero-shot anomaly detection, e.g., \[1,2,3\], is missing.**
>
> We thank the reviewer for pointing us towards these related works. We have updated our related work section and added the works suggested by the reviewer. CLIP-AD \[1\] and ZOC \[2\] are early works in the utilization of CLIP for anomaly/out-of-distribution detection task, and they primarily focus on anomaly classification tasks, whereas AnomalyCLIP jointly optimizes anomaly detection and segmentation tasks, enabling both tasks within a single forward pass during inference. Despite ACR \[3\] addressing detection and segmentation tasks, it requires distinct auxiliary data for specific zero-shot detection tasks. AnomalyCLIP can adapt to various datasets and domains for zero-shot detection by learning object-agnostic prompts only once.
>
> >**References:**
> >1. Philipp Liznerski, Lukas Ruff, Robert A Vandermeulen, Billy Joe Franks, Klaus Robert Muller, and Marius Kloft. Exposing outlier exposure: What can be learned from few, one, and zero outlier images. Transactions on Machine Learning Research, 2022.
> >2. Sepideh Esmaeilpour, Bing Liu, Eric Robertson, and Lei Shu. Zero-shot out-of-distribution detection based on the pretrained model clip. In Proceedings of the AAAI conference on artificial intelligence, 2022
> >3. Aodong Li, Chen Qiu, Marius Kloft, Padhraic Smyth, Maja Rudolph, Stephan Mandt. Zero-Shot Batch-Level Anomaly Detection. Thirty-seventh Conference on Neural Information Processing Systems, 2023.
>
> **Q4: Why is learning two text prompts necessary?**
>
> We have supplemented experiments to investigate the effect of shared prompts in Table 11 in Appendix D. When sharing the learnable word embeddings of $g_n$ and $g_a$, AnomalyCLIP achieves 90.5% AUROC in pixel level and 90.9% in image-level on MVTec AD and 95.0% AUROC in pixel level and 81.5% AUROC in image level on VisA. The results show that AnomalyCLIP without sharing also works well for ZSAD and the efficiency of our object-agnostic prompt learning. However, the shared prompt performs slightly worse than the unshared prompts (used in the original paper). The performance decrease is 0.6%AUROC and 0.6%AUROC in image level on MVTec AD and Visa, and 0.5%AUROC and 0.6%AUROC in pixel level. We believe that the separate learning for these two prompts helps discriminate the generic normality and abnormality because when we share the parameters of $V_i$ and $W_i$, the learned semantics of normal and anomaly may be confused.

---

> > ### Author Response · Authors · 2023-11-22
> > **Response to Reviewer GzuZ (Part 2)**
> >
> > **Q5: How is the performance of the variant with all components except DPAM?**
> >
> > We have added the experiment in Table 7 of Appendix D. When we remove DPAM, the results show a decrease from 91.1% AUROC to 87.9% AUROC in pixel-level performance and from 91.5% AUROC to 80.7% AUROC in image-level performance. This highlights the crucial role played by DPAM in improving AnomalyCLIP's anomaly detection performance. DPAM enhances local visual semantics by modifying the attention mechanism. However, the results indicate that the performance decrease at the image level is more pronounced than that at the pixel level. This discrepancy is attributed to the fact that the total loss places greater emphasis on local context optimization, driven by a larger local loss compared to the case with DPAM. As a result, AnomalyCLIP lacks sufficient optimization for global alignment, leading to a significant decline in image-level performance. Note that the original CLIP (no DPAM) exploits $Q$-$K$ features and suffers from weak segmentation performance.

---

### Official Review · Reviewer_w1Ea · 2023-10-30

**Soundness:** 3 good
**Presentation:** 2 fair
**Contribution:** 2 fair
**Rating:** 5
**Confidence:** 4

**Summary:**

The paper proposes a new method named AnomalyCLIP which deal with the Zero-shot anomaly detection problem. It leverages the CLIP prompt learning techniques to capture the normal/abnormal information within an image regardless of its foreground objects. The experiment results show the effectiveness of their method.

**Strengths:**

1.	Leveraging pre-trained model like CLIP to address anomaly detection is a good direction and an interesting topic.
2.	The paper shows comprehensive and detailed experiments and results which outperforms other baselines.

**Weaknesses:**

1.	The idea and some techniques, like glocal context optimization, are similar to a concurrent work[1]. The author may compare with highly related works.
[1] Gu, Zhaopeng, et al. "AnomalyGPT: Detecting Industrial Anomalies using Large Vision-Language Models." arXiv preprint arXiv:2308.15366 (2023).
2.	The DPAM strategy is confusing. The author claims that the Q-Q, K-K, V-V self-attention suffers from different issues, and V-V self-attention derives the best result. However, these three variants are very similar in the DPAM mechanism. It is better to explain it in the paper and show the results of the original Q-K attention.
3.	I doubt the contribution of some modules. The module ablation results indicate that the T2 object-agnostic text prompts module contributes the most, similar to CoOp method. However, the author employed a different prompt for the CoOp experiment. For instance, the text prompt template for normality is defined as [V1][V2]...[VN][normal][cls], while that for abnormality is [V1][V2]...[VN][anomalous][cls]. This variation in prompt settings doesn't allow for a fair comparison.

**Questions:**

See Weakness section.

---

> ### Author Response · Authors · 2023-11-22
> **Response to Reviewer w1Ea (Part 1)**
>
> Thank you for your review and the insightful comments.
>
> **Q1: The idea and some techniques, like glocal context optimization, are similar to a concurrent work \[1\].**
>
> Thank you for pointing us toward this concurrent work. AnomalyGPT \[1\] is a great work for anomaly detection. Below we summarize the difference between AnomalyGPT and AnomalyCLIP as follows:
>
> 1. The scope is different. AnomalyGPT **eliminates the need for manual threshold adjustments** via LLM when detecting the seen objects during training. However, AnomalyCLIP aims to **improve the model generalization to unseen objects across diverse datasets and domains.**
> 2. We focus on different settings. AnomalyGPT focuses on **unsupervised anomaly detection** and **few-shot anomaly detection**. However, our work aims at **zero-shot anomaly detection**.
> 3. Although both AnomalyGPT and AnomalyCLIP propose to combine local and global loss, the purpose is different. AnomalyGPT uses global loss to **quantify the disparity between the text sequence generated by the model and the target text sequence**, ensuring LLM produces corresponding textual descriptions. However, our proposed global context optimization serves to **align the global visual semantics for anomaly classification.**
> 4. The supervision information is different. Besides the images and text prompts, AnomalyGPT needs **human-crafted text descriptions for different objects to query the LLM during fine-tuning**. AnomalyCLIP just **uses the images and text prompts.**
> 5. We introduce object-agnostic prompt learning to capture the generic normality and abnormality. AnomalyGPT uses human-crafted text prompts to produce textual embeddings.
> 6. We propose DPAM to persevere local visual semantics of the visual encoder without additional training. AnomalyGPT introduces a lightweight image decoder to learn the adaption of local features.
>
> In summary, since these two works have different scopes and focus on different aspects of anomaly detection, and the required supervision information is also different. We can not provide a comparison of the quantitative results between AnomalyGPT and AnomalyCLIP. Instead, we include and discuss AnomalyGPT in Related Work in the revised manuscript, which is marked in orange.
>
> >**Reference:**
> >1. Zhaopeng Gu, Bingke Zhu, Guibo Zhu, Yingying Chen, Ming Tang, and Jinqiao Wang. Anomalygpt: Detecting industrial anomalies using large vision-language models, 2023.
>
> **Q2: The DPAM strategy is confusing. These three variants are very similar in the DPAM mechanism. It is better to explain it in the paper and show the results of the original Q-K attention.**
>
> Thank you for raising this concern. Please refer to **General Response** for a detailed explanation of the DPAM strategy from the attention mechanism. We have also explained the reason why the $V$-$V$ achieves the best performance among DPAM variants there. We have added the response in **Refinement of the local visual space** in Section 3.3 in the updated paper, as well as a more detailed analysis in Appendix C.

---

> > ### Author Response · Authors · 2023-11-22
> > **Response to Reviewer w1Ea (Part 2)**
> >
> > **Q3: I doubt the contribution of some modules.**
> >
> > Thank you for detailed feedback. To demonstrate the effectiveness of each module we propose, we have supplemented the additional module ablation in Table 7 of Appendix D. We test the contribution of one module by removing one module and maintaining the rest module.
> >
> > 1. The effectiveness of DPAM ($T_1$). When we remove DPAM, the results show a decrease from 91.1% AUROC to 87.9% AUROC in pixel-level performance and from 91.5% AUROC to 80.7% AUROC in image-level performance. This performance decline indicates the importance of DPAM, which enhances local visual semantics by modifying the attention mechanism. However, the decrease in performance at the image level is more pronounced than that at the pixel level. This discrepancy is attributed to the fact that the total loss places greater emphasis on local context optimization, driven by a larger local loss compared to the case with DPAM.
> > 2. The effectiveness of object-agnostic prompt learning ($T_2$). Excluding object-agnostic prompt learning makes AnomalyCLIP suffer from the huge performance gap (i.e., 91.1% AUROC to 84.3% AUROC in pixel-level and 91.5% AUROC to 65.6% AUROC in image-level). This performance decline illustrates that the object-agnostic text prompt template plays a significant role in improving the performance of AnomalyCLIP at both pixel and image levels.
> > 3. The effectiveness of textual prompt tuning ($T_3$). When removing textual prompt tuning, the performance of AnomalyCLIP declines from 91.1% AUROC to 90.6% AUROC in pixel-level performance and from 91.5% AUROC to 90.4% AUROC in image-level performance. This demonstrates the importance of adapting original textual space by adding learnable textual tokens in the text encoder.
> > 4. The effectiveness of the integration of multi-layer local visual semantics ($T_4$). When removing multi-layer local visual semantics, the outcomes reveal a decrease from 91.1% AUROC to 90.0% AUROC in pixel-level performance and from 91.5% AUROC to 91.0% AUROC in image-level performance. This performance decline indicates the importance of incorporating multi-layer local visual semantics.
> >
> > **Q4: Similar to CoOp method. However, the author employed a different prompt for the CoOp experiment.**
> >
> > Thank you for your comments. Firstly, the reason for adopting the text prompt templates $[V_1][V_2]...[V_N][\textbf{normal}][\verb'cls']$ and $[V_1][V_2]...[V_N][\textbf{anomalous}][\verb'cls']$ in CoOp is because the common text prompt template used in CLIP for anomaly detection is `A photo of a `**`normal`**` [cls]` and `A photo of an `**`anomalous`**` [cls]`. The distinction between CLIP and CoOp lies in the use of learning prompts. In the CLIP vs. CoOp comparison, we aim to demonstrate the performance difference between fixed and learnable text prompts. Secondly, we have introduced an experiment in Appendix D, where CoOp utilizes the text prompt template suggested by the reviewer, denoted as CoOp$_2$, with the template $[V_1][V_2]...[V_N][\verb'cls']$ and $[V_1][V_2]...[V_N][\textbf{damaged}][\verb'cls']$ representing normality and abnormality, respectively. The original CoOp version is referred to as CoOp$_1$. We compare the performance of CoOp$_1$ and CoOp$_2$ across different datasets and domains, presenting the results in Table 9 and Table 10. CoOp$_1$ and CoOp$_2$ have very similar performance, though the text prompt varies. Additionally, we explore $[V_1][V_2]...[V_N][\textbf{anomalous}][\verb'object']$ for AnomalyCLIP in Table 16 and Table 17 of Appendix D. These results suggest that whether using the same prompt template or not, we observe a similar superiority of AnomalyCLIP over CoOp.

---

### Official Review · Reviewer_xREe · 2023-11-01

**Soundness:** 3 good
**Presentation:** 3 good
**Contribution:** 3 good
**Rating:** 8
**Confidence:** 3

**Summary:**

In the paper, the authors address Zero-shot anomaly detection (ZSAD) where training data is unavailable. They introduce AnomalyCLIP, which improves CLIP's generalization for ZSAD by using object-agnostic prompt learning. This allows for generic recognition of normality and abnormality across diverse images. AnomalyCLIP also incorporates both global and local anomaly contexts for optimization. Tested on 17 datasets, AnomalyCLIP demonstrates superior ZSAD performance.

**Strengths:**

The paper introducing AnomalyCLIP for Zero-shot anomaly detection (ZSAD) stands out for its originality, exemplified by its novel approach of using object-agnostic text prompts for anomaly detection, a creative departure from traditional methods. The quality of the work is underscored by its robust methodology and extensive validation across 17 diverse datasets. The authors effectively communicate their ideas with clarity, making the complex concepts accessible. Significantly, AnomalyCLIP's ability to generalize across various domains without needing prompt redesign, particularly in industrial and medical settings, marks it as a notable advancement in the field of anomaly detection.

**Weaknesses:**

A weakness in the paper is the unexplained initial use of the term "glocal." Clarifying this key term when first mentioned would improve understanding and clarity.

**Questions:**

See the weaknesses.

---

> ### Author Response · Authors · 2023-11-22
> **Response to Reviewer xREe**
>
> We thank the reviewer for the positive and encouraging review. When initially using the term 'glocal' in Section Introduction, we have added the explanation in the revised manuscript: "We then introduce a novel ZSAD approach, called AnomalyCLIP, in which we utilize an object-agnostic prompt template and a glocal abnormality loss function (**i.e., a combination of global and local loss**) to learn the generic abnormality and normality prompts using auxiliary data.\"

---

### Official Review · Reviewer_NCKk · 2023-11-01

**Soundness:** 2 fair
**Presentation:** 2 fair
**Contribution:** 2 fair
**Rating:** 5
**Confidence:** 3

**Summary:**

This paper propose a zero-shot anomaly detection (ZSAD) approach using CLIP, named AnomalyCLIP. Different from the previous ZSAD methods, AnomalyCLIP attempts to learn object-agnostic text prompts of normality and abnormality to segment abnormal part. To reach this, the authors leverages textual prompt tunning and DPAM technologies. Extensive evaluations conducted on 17 publicly datasets demonstrates the effectiveness of the proposed approach.

**Strengths:**

1. The empirical evaluations are extensive covering 17 diverse benchmarks.

2. The proposed approach attempts to learn class-agnostic prompts which seems to contradict the pretrained CLIP that is mostly used to classify semantic objects. The authors successfully mitigate this issue by several technical modules, e.g. object-agnostic text prompt design and DPAM layer.

**Weaknesses:**

1. The prompt template might be too restrictive. "damaged [cls]" may not well represent all types of anomalies. For example, if a component is missing or applying the method to other domains than defect identification the proposed prompt template may not work well.

2. According to Figure 2, the pipeline needs to feed the same images into two visual encoders, this would introduce additional computation overhead.

3. The proposed DPAM layer lacks a theoretical basis. It is not clear why replacing Q-K attention map with V-V attention map without updating any visual encoder's parameters into CLIP would improve the performance.

4. The paper has not discussed why the proposed approach is able to learning class-agnostic feature from the pretrained semantic CLIP as the CLIP is trained to be sensitive to semantic classes.

**Questions:**

It is important to discuss why the "damaged [cls]" is suitable for more diverse anomaly detection scenarios.

Explanations on the effectiveness of DPAM and computation overhead are necessary.

---

> ### Author Response · Authors · 2023-11-22
> **Response to Reviewer NCKk (Part 1)**
>
> Thank you very much for your review and the insightful comments.
>
> **Q1: The prompt template might be too restrictive. "damaged \[cls\]\" may not well represent all types of anomalies.**
>
> We are not sure about whether the reviewer refers to the object-agnostic text prompt "damaged **\[object\]**\" (where `object` is a fixed token like `damaged`) or the object-aware text prompt "damaged **\[cls\]**\" (where `cls` is a variable token that will be replaced with a specific class name in the algorithm implementation).In our submission, AnomalyCLIP uses the former one. Nevertheless, we posit that the reviewer's main concern is on the limited scope of the token "damaged\" may have for anomaly detection.
>
> Since the pattern of anomaly is typically unknown and diverse, it is practically difficult to list all possible anomaly types. Thus, we agree that the prompt "`damaged [object]`\" can be restrictive. To remedy this problem, during the fine-tuning process, we provide pixel-level anomaly ground truth with diverse anomaly types for training our ZDAD. This way helps substantially extend the scope of the prompt template "`damaged [object]`\". It indicates in the linguistic semantic of the token "`damaged`\", incorporating fine-grained anomaly semantics of diverse anomaly types. This intuition is verified by our ablation study for the prompt template in Table 16 and Table 17 of Appendix D, in which AnomalyCLIP shows stable performance across different datasets when replacing "`damaged`\" with other similar tokens: "`anomalous`\", "`flawed`\", "`defective`\", and "`blemished`\".
>
> As for the reason why such a prompt template can be applied to other domains, we believe that the learnable part of the prompt template is optimized to focus on the anomaly regions regardless of its foreground object, which allows our model to capture generic object-agnostic, or even domain-agnostic normality and abnormality. Further, compared to object-aware text prompt templates (i.e., "`damaged [cls]`\"), excluding the object semantics from text prompt templates allows learnable text prompt templates to focus on capturing the characteristics of anomalies themselves, rather than the objects. Additionally, the strong generalization of the visual encoder is also important because it can project images from different domains into the same embedding space. Therefore, AnomalyCLIP can detect medical datasets even when fine-tuned on the industrial defect dataset, as shown in Table 2.
>
> **Q2: Explanations on the effectiveness of DPAM and computation overhead are necessary.**
>
> We provide a detailed explanation of DPAM and attempt to analyze the effectiveness of the attention mechanism in **General Response**. **We would like to clarify that AnomalyCLIP does not need to feed the images into two visual encoders**. In Figure 2, we plot two paths to intuitively provide the computation difference between global and local visual embedding. **Without introducing additional parameters, we just create two paths when computing the attention map (DPAM), and the rest process shares the same visual encoder.** As for computation overhead, we have supplemented the assessment of the time taken during training (training time per epoch) and the inference speed (frames per second, FPS). For a fair comparison, all experiments are conducted in a single 3090 NVIDIA RTX 3090 24GB GPU, and the GPU is kept free before evaluation. In Table 8, AnomalyCLIP takes 13.71 min per epoch on MVTec AD (The total number of samples is 1725) and only requires a total of 15 epochs for the whole fine-tuning. Once AnomalyCLIP finishes fine-tuning, AnomalyCLIP can be applied to different datasets and domains without additional training. We also compare AnomalyCLIP with other baselines that need auxiliary data (i.e., CoOp and VAND). The minimum training time per epoch is 12.25 min of CoOp, and hence the training time taken is similar for fine-tuning methods. As for inference speed, CLIP achieves the 13.23 FPS. However, it suffers from weak detection performance. Although WinCLIP achieves better performance, WinCLIP has only 1.2 FPS because it needs multiple forward image patches to derive the segmentation. AnomalyCLIP outperforms WinCLIP and obtains an FPS of 8.92.
>
> We also evaluated the computation overhead of DPAM separately. In Table 8, without DPAM, AnomalyCLIP takes 12.98 min to train per epoch. Compared to the 13.71 min for AnomalyCLIP with DPAM, we observe that introducing DPAM does not significantly increase the time complexity. This is attributed to the fact that DPAM only creates the two paths during the computation of the attention map and is frozen during fine-tuning, thereby avoiding the computationally expensive process of gradient computation. Meanwhile, DPAM also does not result in large computation overhead during inference: AnomalyCLIP w/ DPAM gets 8.92 FPS vs. 10.21 FPS for w/o using DPAM.

---

> > ### Author Response · Authors · 2023-11-22
> > **Response to Reviewer NCKk (Part 2)**
> >
> > **Q3: Why the proposed approach is able to learn class-agnostic features from the pretrained semantic CLIP as the CLIP is trained to be sensitive to semantic classes.**
> >
> > Thank you for your insightful comments. To mitigate the sensitivity to object semantics in CLIP, AnomalyCLIP proposes three modules to adapt the original textual and visual embeddings to object-agnostic anomaly semantics. Firstly, we propose object-agnostic prompt learning to exclude the object semantics from text prompt templates and allow learnable text prompt templates to focus on capturing the characteristics of anomalies themselves, rather than the objects. This empowers AnomalyCLIP to generate the appropriate embeddings in the textual space. However, the object-agnostic embedding is not only decided by the text prompt but also depends on the appropriate textual space. Therefore, we also introduce learnable textual tokens in the textual encoder to adapt the textual space. Besides the textual encoder, we also propose DPAM to improve the local visual embeddings to facilitate the learning of object-agnostic features. Finally, during fine-tuning, AnomalyCLIP is optimized by glocal context optimization to focus on image-level and pixel-level anomalies across diverse anomaly types regardless of object semantics. This process further enables textual embeddings to be more generic and capable of identifying anomalies across diverse objects and different domains.

---

### Author Response · Authors · 2023-11-22
**General Response (Part 1)**

Dear Revewers and ACs,

We very much appreciate the detailed review, and positive and constructive comments from all of the reviewers. While we responded to each of the reviewer comments individually, we also provide a response to the common concern below:

- Why the proposed DPAM is effective in improving local visual semantics of CLIP (Reviewers **NCKk**, **w1Ea**, and **vcXy**)?

Since the visual encoder of CLIP is originally pre-trained to align global object semantics, such as cat and dog, the contrastive loss used in CLIP makes the visual encoder produce a representative global embedding for recognizing semantic classes. Through the self-attention mechanism, the attention map in the visual encoder focuses on the specific tokens highlighted within the red rectangle in Figure 3(b). Although these tokens may contribute to global object recognition, they disrupt the local visual semantics, which directly hinders the effective learning of the fine-grained abnormality in our object-agnostic text prompts. For segmentation purposes, it's crucial for the visual feature map to emphasize the surrounding context to capture more local visual semantics.

Formally, let $a_{ij}$ be an attention score in the attention score matrix, where $i, j \in [1, h \times w]$, then the i-th output of $Q$-$K$ attention can be written as:
$$ \quad \quad \quad \quad \quad \quad \quad \quad \quad \quad \quad Attention( {Q}, {K}, {V})\_i = softmax\left(\frac{q_iK^{\top}}{\sqrt{D}}\right)V = \frac{\sum\limits_{j=1}^n a_{ij} v_j}{\sum\limits_{j=1}^n a_{ij}} \quad \quad a_{ij} =  e^{  \frac{q_i k_j^{\top}}{\sqrt{D}}}.$$
$Attention( {Q}, {K}, {V})\_i$ can be regarded as the weighted average of $v_j$ using $a_{ij}$ as the weight. Assuming that the original attention map focuses on the specific tokens at index $m$, it is clear that $q_i$ only produces the large attention score with $k_m$ in all $k_j$. Therefore, $a_{im}$ is the largest score among other $a_{ij}$ so $Attention({Q}, {K}, {V})\_i$ is dominated by $v_m$, which causes the local visual embedding at index $i$ to be disturbed by the local visual embedding at index $m$. In Figure 3(b), the attention score map presents vertical activation and suggests that every $q_i$ produces a large attention score with $k_m$. In such a case, each $Attention({Q}, {K}, {V})\_i$ is dominated by $v_m$ and results in weak anomaly segmentation in Figure 3(b) even though $v_m$ may be important for original class recognition. Some prior studies \[1\] \[2\] use an additional decoder to recover the local visual semantics. In this paper, we directly use local visual embeddings for segmentation and point out that an ideal attention map for local visual semantics should exhibit a more pronounced diagonal pattern. For this purpose, DPAM is proposed to replace the original $Q$-$K$ attention with analogous components, including $Q$-$Q$, $K$-$K$, and $V$-$V$ self-attention. Therefore, $a_{ij}$ is changed into:
$$ \quad \quad \quad \quad \quad \quad \quad \quad \quad \quad \quad \quad \quad \quad \quad  a^{qq}\_{ij} =  e^{ \frac{q_i q_j^{\top}}{\sqrt{D}}}, \quad  a^{kk}\_{ij} =  e^{ \frac{k_i k_j^{\top}}{\sqrt{D}}}, \quad a^{vv}\_{ij} =  e^{ \frac{v_i v_j^{\top}}{\sqrt{D}}}.$$
    This modification ensures that $q_i$, $k_i$, and $v_i$ hold significant weight in forming $Attention({Q}, {Q}, {V})\_i$, $Attention({K}, {K}, {V})\_i$, and $Attention({V}, {V}, {V})\_i$, thereby preserving local visual semantics. As a result, the produced attention maps exhibit a more diagonal prominence compared to the original Q-K attention, leading to improved performance in anomaly segmentation, as shown in Figures 3(b), (c), and (e). However, since $Q$ and $K$ consist of the original attention map, important tokens at index $m$ for class recognition within themselves may also produce relatively large scores ($a_{im}$) (e.g., $q_i$ has strong relevance with $q_m$ besides $q_i$) to disturb $Attention({Q}, {Q}, {V})\_i$ and $Attention({K}, {K}, {V})\_i$ in Figure 3(c) and Figure 3(d). In contrast to $Q$-$Q$ and $K$-$K$, $V$-$V$ does not participate in computing the original attention map, reducing the unexpected bias to different tokens in $V$ for the purpose of anomaly segmentation. Therefore, $v_i$ does not produce a large weight ($a_{ij}$) with $v_j$ and generates a larger weight ($a_{ii}$) to form $Attention({V}, {V}, {V})\_i$, preserving more information of $v_i$ and experiencing diagonally prominent attention map (minimal disturbance), as depicted in Figure 3(e). This is the reason why $V$-$V$ achieves the best results.

While DPAM is part of AnomalyCLIP, we would like to clarify that the key innovations in our work lie in (i) the revelation of the effectiveness of object-agnostic normality and abnormality text prompt learning for ZSAD and (ii) the proposed approach in AnomalyCLIP for such text prompt learning, consisting of prompt template design, global and local context optimization, textual and local visual space refinement.

---

> ### Author Response · Authors · 2023-11-22
> **General Response (Part 2)**
>
> We also provide a brief summary of the main changes made to the revised version of the paper in response to the reviews:
>
> - As recommended by Reviewer **NCKk**, we have provided a more detailed analysis of DPAM and examined the computation overhead.
> - To address the concerns raised by Reviewers **w1Ea**, **GzuZ**, and **vcXy**, we have added the suggested module ablation study by systematically removing each module to highlight its effectiveness, particularly in the case of DPAM.
> - In response to the feedback from Reviewer **GzuZ**, we have included an analysis of successful cases and potential failures.
> - Ablation studies have been conducted on various aspects, including the text prompt template of CoOp (Reviewer **w1Ea**), the unshared text prompt (Reviewer **GzuZ**), local visual features ensemble (Reviewer **vcXy**), the backbone of CLIP (Reviewer **5wSC**), and local loss (Reviewer **5wSC**).
> - We have also addressed the feedback from Reviewers **w1Ea** and **GzuZ** by incorporating important references that were previously missing.
>
> We have uploaded the revised paper which includes additional experiments and various clarifications to address the feedback from the reviewers. For ease of review, we highlight the revised text in orange. For other questions raised by the reviewers, please see our response to individual questions and concerns below each review.
>
> >**References:**
> >1. Yongming Rao, Wenliang Zhao, Guangyi Chen, Yansong Tang, Zheng Zhu, Guan Huang, Jie Zhou, and Jiwen Lu. Denseclip: Language-guided dense prediction with context-aware prompting. In Proceedings of the IEEE/CVF Conference on Computer Vision and Pattern Recognition, pp. 18082--18091, 2022.
> >2. Zhaopeng Gu, Bingke Zhu, Guibo Zhu, Yingying Chen, Ming Tang, and Jinqiao Wang. Anomalygpt: Detecting industrial anomalies using large vision-language models, 2023.

---

### Meta-Review · Area_Chair_v57G · 2023-12-04

**Metareview:**

This paper proposes a zero-shot anomaly detection framework using CLIP, object-agnostic text prompts with learnable prompt tokens, and a modified attention mechanism. The proposed method achieves state-SOTA results on various datasets.

Overall, three out of six reviewers provided positive reviews and considered the proposed probabilistic framework for zero-shot anomaly detection novel, with the method achieving SOTA results. The authors addressed the comments from these reviewers during the rebuttal process, which I found satisfactory; however, none of the three reviewers responded to the authors' rebuttal.

Three reviewers, namely Reviewer NCKk, Reviewer w1Ea, and Reviewer vcXy, provided slightly negative reviews. While acknowledging the SOTA results and finding the proposed method interesting with potential impact on the zero-shot anomaly detection community, their main concerns focused on the novelty of the DPAM module, and the contributions of the different modules proposed and used by the authors. The authors responded to these concerns with detailed explanations about the DPAM module and empirical justifications for the contributions of different modules via ablation tests. None of the three reviewers responded to the authors' rebuttal; however, I found the authors' responses satisfactory.

Considering the above discussion and the rebuttal/changes made to the paper, I recommend acceptance. I believe the idea is novel, and the experiments demonstrate state-of-the-art results, which will have an impact on the machine learning community specializing in zero-shot anomaly detection.

**Justification For Why Not Higher Score:**

The rebuttal didn't considerably alter the reviewers' rating.

**Justification For Why Not Lower Score:**

A lower score would typically result in the rejection of the paper; however, this paper holds the potential to make an impact on the ML community focusing on zero-shot anomaly detection.

---

### Decision · Program_Chairs · 2024-01-16

Accept (poster)